# FreSh: Frequency Shifting for Accelerated Neural Representation Learning

**Adam Kania[1], Marko Mihajlovic[2], Sergey Prokudin[2,3], Jacek Tabor[1], Przemysław Spurek[1,4]**
Faculty of Mathematics and Computer Science, Jagiellonian University, Krakow, Poland[1];
ETH Zurich[2]; Balgrist University Hospital[3]; IDEAS NCBR[4]

## Abstract

Implicit Neural Representations (INRs) have recently gained attention as a powerful approach for continuously representing signals such as images, videos, and 3D shapes using multilayer perceptrons (MLPs). However, MLPs are known to exhibit a low-frequency bias, limiting their ability to capture high-frequency details accurately. This limitation is typically addressed by incorporating high-frequency input embeddings or specialized activation layers. In this work, we demonstrate that these embeddings and activations are often configured with hyperparameters that perform well on average but are suboptimal for specific input signals under consideration, necessitating a costly grid search to identify optimal settings. Our key observation is that the initial frequency spectrum of an *untrained* model's output correlates strongly with the model's eventual performance on a given target signal. Leveraging this insight, we propose *frequency shifting* (or FreSh), a method that selects embedding hyperparameters to align the frequency spectrum of the model's initial output with that of the target signal. We show that this simple initialization technique improves performance across various neural representation methods and tasks, achieving results comparable to extensive hyperparameter sweeps but with only marginal computational overhead compared to training a single model with default hyperparameters. The code is available at: https://github.com/gmum/FreSh/

## 1 Introduction

Implicit Neural Representations (INRs) are advancing computer graphics research by integrating classical algorithms with continuous signal representations. They have been successfully applied in signal representation and inverse problems, with notable applications in neural rendering, compression, and 2D and 3D signal reconstruction (Xie et al., 2022).

INRs primarily rely on multilayer perceptrons (MLPs), making them susceptible to *spectral bias*, which refers to the slower convergence of MLPs when approximating high-frequency components of the target signal (Rahaman et al., 2019). Although spectral bias can benefit generalization (Ronen et al., 2019), it can also hinder performance (Gorji et al., 2023; Rahaman et al., 2019), especially in scenarios that require high precision of signal reconstruction, such as the training of INRs. This led to the development of numerous architectures aimed at overcoming *spectral bias* and its resulting capacity constraints by increasing the frequencies present in the input signal at the first layer of the model (embedding layer), e.g., through frequency-changing activation functions (Sitzmann et al., 2020b; Liu et al., 2024; Tancik et al., 2020) or auxiliary data structures (Chan et al., 2022; Müller et al., 2022).

The frequency of such embedding layers is typically controlled by hyperparameters whose configuration can significantly affect performance (see Figure 1). In section 3 we show that default hyperparameter values can hinder performance and lead to blurry reconstructions. Improving performance is possible by using parameters sweeps, however, optimizing embedding parameters by training multiple models introduces significant overhead and is not feasible in practice.

The high computational cost of performing a parameter sweep comes from training each model. Instead of training, we approximate model performance using the Fourier transform and the Wasserstein distance (Wasserstein, 1969), which was approximately 20 times faster than grid search in our

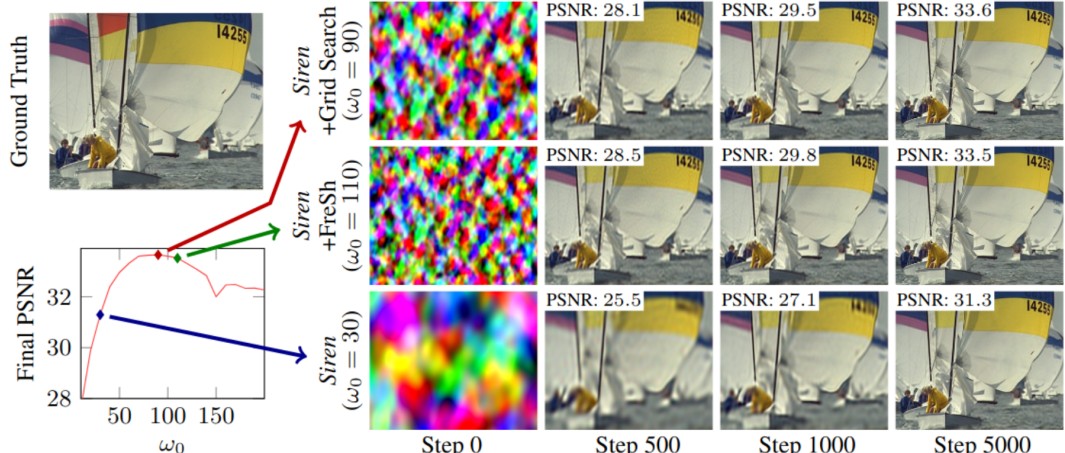

Figure 1: The configuration of embeddings is crucial for the convergence speed. We train *Siren* with various embedding configurations ($\omega_0 \in [10, 200]$) for 5k steps on a Kodak image (top-left). The best grid-search found model ($\omega_0 = 90$), the FreSh configuration ($\omega_0 = 110$) and the baseline ($\omega_0 = 30$) are marked with diamonds (bottom-left). The optimal and FreSh configurations (top and middle rows) lead to sharper details, such as the number on the sail, compared with the baseline (bottom row). Even though *Siren* uses a frequency embedding, the baseline is blurry due to low frequency bias. Note how the sizes of uniformly colored areas in the output at step 0 indicate the size of image features the network can easily learn - this observation is pivotal for FreSh.

experiments (see Table 4). Our method, dubbed FreSh (**Fre**quency **Sh**ifting for Accelerated Neural Representation Learning), selects the embedding configuration where the frequency distribution of the model's output is close to that of the target signal. This shift in the model's frequency distribution results in better signal modeling (see Figure 1). We validate our approach experimentally, demonstrating improved quality in representation tasks such as image and video overfitting and in an inverse problem, specifically 3D shape modeling with NeRF. We show that these improvements result from a more accurate approximation of all frequencies (see Figure 6).

In existing INR studies (Sitzmann et al., 2020b; Tancik et al., 2020; Saragadam et al., 2023; Liu et al., 2024), new architectures are often introduced with minimal attention to simplifying the costly process of hyperparameter selection, even though it is crucial for achieving the best performance. Our framework is the first to address this gap by leveraging frequency information to guide the configuration process for embeddings of existing INR models, with the potential to easily extend to future architectures. Our key contributions are:

- We develop a technique for comparing frequency contents of images based on the Discrete Fourier Transform and the Wasserstein distance.

- We introduce FreSh (**Fre**quency **Sh**ifting for Accelerated Neural Representation Learning) - a model agnostic method for configuring coordinate embeddings for better performance, that can be easily applied using a provided script. FreSh works by adjusting the capacity of an INR to model all target signal's frequencies.

- We achieve state-of-the-art results in image and video approximation, as well as 3D shape reconstruction (NeRF), using a fraction of the compute of a conventional grid search.

## 2 RELATED WORK

**INRs** are neural models used for signal representation that received considerable attention in research (Tewari et al., 2022) and have been applied in various domains, including representation of images (Klocek et al., 2019), videos (Chen et al., 2022b), and 3D shapes (Park et al., 2019). Notable applications include 3D shape reconstruction (Mildenhall et al., 2021), robotics (Wang et al., 2021b; Lin et al., 2021), and compression (Lu et al., 2021; Takikawa et al., 2021). INR architectures are often simple, consisting of a single MLP, with improvements focusing on embedding layers (Chen et al., 2022a; Müller et al., 2022), activations (Sitzmann et al., 2020b), rendering techniques (Barron et al., 2021; 2023), and regularization methods (Yang et al., 2023).

**Spectral bias** is a phenomenon observed in MLPs describing their preference for learning low-frequency functions and ignoring high-frequency noise (Rahaman et al., 2019; Ronen et al., 2019; Xu, 2018), which helps explain the remarkable generalization properties of deep models. Nevertheless, this low-frequency bias hurts model performance when high-frequency components of the signal are informative (Gorji et al., 2023). A common way of overcoming this bias in INRs is by introducing high-frequency embeddings (Sitzmann et al., 2020b; Mildenhall et al., 2021; Müller et al., 2022; Liu et al., 2024) that change the space over which the MLP operates, thus modifying the frequencies of the target function. However, the effectiveness of such approaches is limited as optimal hyperparameter configurations have to be found by trial and error for each target signal.

**Positional encodings** are a broad class of functions that map coordinates into a high dimensional space through a function of adjustable frequency. Their particularly prominent use is in Transformers (Vaswani et al., 2017), but they are also crucial for INRs. One of their first applications was NeRF (Mildenhall et al., 2021), a model for 3D scene reconstruction from a set of posed 2D images. NeRF embeddings are not stable in sparse settings due to its usage of very high frequencies (Yang et al., 2023), and its axis-alignment makes reconstruction quality rotation-dependent (Tancik et al., 2020). Despite its drawbacks, it is frequently used (Barron et al., 2021; Pumarola et al., 2021; Barron et al., 2023), which could be due to its low sensitivity to hyperparameters compared to alternatives. These alternatives include various activation functions (Sitzmann et al., 2020b; Saragadam et al., 2023) and a direction-invariant version of NeRF using Fourier features (Tancik et al., 2020).

**Activation functions** are a popular method of improving the capacity of INRs, where the first layer acts as a positional embedding. A well-known example is *Siren* (Sitzmann et al., 2020b), which uses a sine activation to increase signal frequencies in the model's first layer. Architectures that generalize this approach utilize the Gabor wavelet (Saragadam et al., 2023), non-periodic functions (Ramasinghe & Lucey, 2022), and variable-periodic functions (Liu et al., 2024). In this work, we particularly focus on *Siren* due to its popularity, but we also address other activations.

**Auxiliary data structures** are used in neural scene representation to associate fragments of the scene with a feature vector of trainable parameters, trading a larger memory footprint for smaller computational costs. However, due to the extremely small MLPs used, these approaches can lose some of global reasoning and implicit regularization (Neyshabur et al., 2014; Goodfellow et al., 2016) capabilities of neural models. As directly storing a fine grid of features would be prohibitively expensive, practical approaches use low-rank approximations (Chen et al., 2022a), 2D feature maps (Chan et al., 2022) and hash tables (Müller et al., 2022) to reduce the memory cost. In such settings spectral bias is not avoided, and the resolution of the voxel grid needs to be tuned for each scene. Although our main goal is to improve pure neural network-based solutions, we also verify on the recent architecture from Müller et al. (2022) that FreSh is applicable to grid-based approaches.

**ResFields** is a novel framework for INRs that improves their capacity for representing complex signals by incorporating temporal residual layers into MLPs (Mihajlovic et al., 2024). By modifying network weights with a time-dependent component represented as a factorized matrix, it increases the performance with only a small impact on parameter count and inference speed. We use ResFields to improve current state-of-the-art results on video representation.

**Initialization schemes for INRs** have been extensively studied to accelerate training convergence. IGR (Gropp et al., 2020) introduced an implicit geometric initialization to speed up learning 3D shapes, while others (Rajeswaran et al., 2019; Sitzmann et al., 2020a; Wang et al., 2021a; Tancik et al., 2021) leveraged data-driven meta-learning approaches (Finn et al., 2017; Nichol, 2018) for learning implicit fields. In contrast to these methods, which rely on computationally expensive pre-training or hand-crafted priors, our approach avoids such requirements, offering a more efficient alternative. Moreover, novel initialization methods such as the ones proposed by Saratchandran et al. (2024) are largely orthogonal with our method.

**Regularization strategies** can be applied to stabilize the training of NeRF-based models, especially in sparse settings (Yang et al., 2023). As our interest is in hyperparameter selection, we do not test the effects of regularization on final results. Moreover, some regularization methods (Yang et al., 2023) were developed only for NeRF-like embeddings and would have to be generalized for a fair comparison. It is also worth noting that the Wasserstein distance has been already applied to improve INRs (Ramasinghe et al., 2024) through regularization, which is different from our application.

Table 1: Comparison between the default, optimal, and FreSh configurations of *Siren* during 15,000-step training on various datasets (Chest X-Ray, FFHQ-1024, FFHQ-wild, Kodak, Wiki Art), using one image per dataset. We provide the PSNR scores and the total training time, which includes $\omega_0$ selection. The default value of $\omega_0$ leads to suboptimal results, but improvements can be achieved by using an optimal configuration found through a hyperparameter sweep ($\omega_0 \in \{30, 40, ..., 140\}$). This process involves training multiple models, making fixing $\omega_0$ a common strategy. FreSh achieves similar improvements to grid search while being an order of magnitude faster and comparable in time with a plain *Siren*. Results are averaged over three seeds. The grid search was performed once, and the best configuration was re-evaluated using different seeds.

| PSNR $\uparrow$ | Step | | | | Training |
|---|---|---|---|---|---|
| | 500 | 1000 | 5000 | 15000 | time (h) $\downarrow$ |
| *Siren* ($\omega_0 = 30$) | 25.26 $\pm 0.03$ | 26.10 $\pm 0.04$ | 29.13 $\pm 0.06$ | 31.18 $\pm 0.02$ | 2.66$\pm > 0.01$ |
| *Siren* + Grid Search | 27.36 $\pm 0.06$ | 28.39 $\pm 0.03$ | 30.51 $\pm 0.03$ | 32.02 $\pm 0.02$ | 32.69$\pm 0.16$ |
| *Siren* + FreSh | 27.11 $\pm 0.03$ | 28.16 $\pm 0.02$ | 30.32 $\pm 0.03$ | 31.81 $\pm 0.01$ | 2.82$\pm 0.01$ |

## 3 MOTIVATION

In this section, we show that the training of an INR highly depends on configuring the embedding in a way that aligns its frequencies with the target signal. This leaves practitioners with two options, either using a suboptimal, default embedding configuration or finding a well performing configuration through a costly parameter sweep.

We illustrate the impact of proper hyperparameter selection in an image representation task in Figure 1 using the *Siren* model, which uses an input-scaling parameter $\omega_0$ to increase embedding frequencies. We compare a default, unaligned with the target signal configuration of *Siren* ($\omega_0 = 30$) to an aligned configuration ($\omega_0 = 90$) selected through a parameter sweep over $\omega_0 \in \{30, \dots, 140\}$. The aligned configuration speeds up training by employing frequencies three times greater than the baseline, achieving sharp details while the baseline is blurry. Note how the sizes of uniformly colored areas in the output at step 0 indicate the size of features the network can easily learn - this observation is pivotal for FreSh. We perform a similar investigation on 5 images, each from a different dataset (see section 5), reporting the results in Table 1. We find that optimizing $\omega_0$ always improves the baseline results, with specific best values of $\omega_0$ depending on the target signal. The failure of SGD to optimize the embedding layer (see Appendix A) necessitates hyperparameter sweeps, as a one-size-fits-all solution will inevitably be suboptimal for some target signals. Our goal is to perform this search and enhance INR performance while avoiding the high computational cost of repeated model re-training.

## 4 FRESH

In this section, we discuss how to compare the frequency contents of images and introduce FreSh, a computationally efficient method for initializing frequency embeddings that biases the model towards the frequencies present in the target signal.

### 4.1 PRELIMINARIES

This section discusses important theoretical concepts relevant to our study. INR architectures are discussed in the Appendix, with the exception of the *Siren* model (Sitzmann et al., 2020b), which we use as a high-level example to illustrate how similar approaches work. We provide a list of embedding hyperparameters from each model used in our study in Table 2.

***Siren*** (Sitzmann et al., 2020b) addresses spectral bias by mapping its inputs, $\mathbf{x} \in \mathbb{R}^d$, through a high-frequency embedding, given as

$$\gamma_S(\mathbf{x}) = \sin(\omega_0 W \mathbf{x} + \mathbf{b}), \tag{1}$$

where $W \sim \mathcal{U}[-\frac{1}{d}, \frac{1}{d}]$ are the weights of the layer and $\mathbf{b}$ is bias. The scaling parameter $\omega_0$ controls the frequency magnitudes of this embedding and the authors (Sitzmann et al., 2020b) recommend

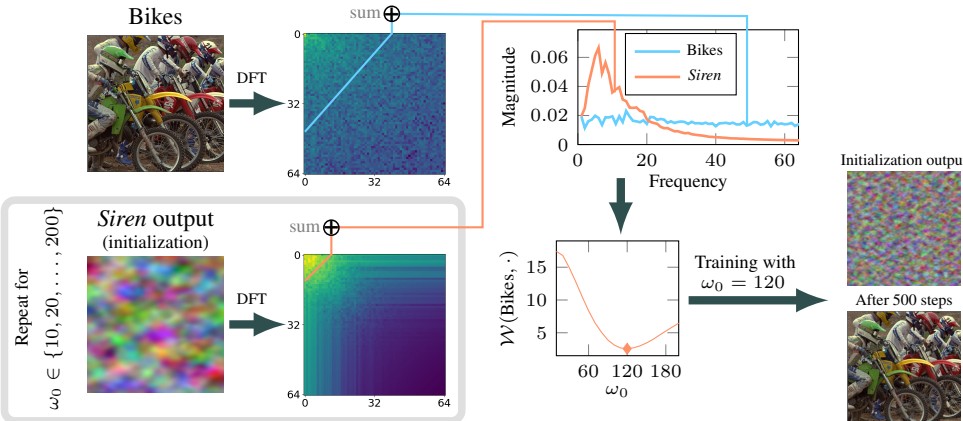

Figure 2: Example workflow of FreSh when applied to *Siren* and a high-frequency Kodak image. The image and outputs from various model configurations undergo a Discrete Fourier Transform (left). The Fourier coefficients of the same degree are then summed to produce the image spectrum (top-right). The model spectra are compared with the dataset spectrum using the Wasserstein distance $\mathcal{W}$ (bottom-middle), with only the configuration at the global minimum, highlighted by a diamond, used for training (bottom-right). Note that the Wasserstein distance follows a smooth trend with a distinct global minimum, indicating stable and predictable behavior. For additional examples of model spectra, see Figure 8.

setting $\omega_0 = 30$, but adjustments are needed to reach optimal performance (Saragadam et al., 2023). *Siren* was specifically designed so that the first layer is the sole contributor to the frequency increase inside the model, which results in the frequency distribution (spectrum) of this model being very concise.

**Other models** adopt a similar approach to *Siren* and control embedding frequencies through a hyperparameter, with higher values corresponding to higher frequencies. Notable models that employ this strategy include NeRF (Mildenhall et al., 2021), *Fourier* (Tancik et al., 2020), *Finer* (Liu et al., 2024), and Hashgrid (Müller et al., 2022). While most models use embedding frequencies that are similar to what is present in the dataset and not excessively high, NeRF employs frequencies that often surpass those found in the dataset, which makes it incompatible with our comparison-based method. Similarly, *Wire* (Saragadam et al., 2023) uses very high frequencies and increases frequencies at each hidden layer, not just at the

Table 2: Embedding hyperparameters of architectures used in our study. In all models increasing the hyperparameter value results in higher embedding frequencies.

| Model | Frequency Parameter | Description |
|---|---|---|
| *Fourier* | $\sigma$ | Weight variance |
| *Siren* | $\omega_0$ | Scaling parameter |
| *Finer* | $\omega$ | Scaling parameters |
| *Finer* | $k$ | Bias range |
| Hashgrid | $N_{max}$ | Grid resolution |

embedding layer. This makes it incompatible with FreSh. As such, we provide results for NeRF and *Wire* as a reference, but do not optimize them with FreSh. The approach of *Finer* differs from other models by employing two hyperparameters, though they largely serve the same purpose, raising questions about the necessity of both. We explore two scenarios for *Finer*: in each scenario, we optimize only one of the parameters.

**Discrete Fourier Transform** (DFT) of an image $A \in \mathbb{R}^{C \times N \times N}$ describes its frequencies based on both their magnitude and direction. For example, $\sin(x_0)$ and $\sin(x_1)$ represent the same frequency magnitude but different directions. We denote the DFT of the $c$-th channel of $A$ as $\mathcal{F}(A_c) \in \mathbb{R}^{N \times N}$, where the element $\mathcal{F}_{j,k}(A_c)$ is defined as

$$\mathcal{F}_{j,k}(A_c) = \sum_{m=0}^{N-1} e^{-i2\pi \frac{jm}{N}} \sum_{n=0}^{N-1} e^{-i2\pi \frac{kn}{N}} A_{c,m,n}. \qquad (2)$$

Since DFT depends on direction, it is not invariant to transposition, i.e., $\mathcal{F}(A_c) \neq \mathcal{F}(A_c^T)$.

Direction dependence makes DFT different from embeddings used in INR models, which treat all inputs symmetrically. Consequently, DFT has an unnecessarily complex structure (a matrix instead

of a vector), capturing differences, such as image transposition, that are irrelevant to our application. To make the DFT more suitable for our purposes, we reduce it to a **spectrum** vector $\mathcal{S}(A) = (\mathcal{S}(A, 1), \ldots, \mathcal{S}(A, N-1))$ which removes direction dependence of DFT ($\mathcal{S}(A) = \mathcal{S}(A^T)$), for examples, see Figure 2. We achieve this by summing together elements of DFT that represent the same frequency, but not direction, meaning the elements where DFT indices sum to the same number. This is well illustrated by transposing an image, as $\mathcal{F}_{i,j}(A_c^T)$ is the same as $\mathcal{F}_{j,i}(A_c)$. Specifically, the elements of the spectrum vector are created by summing along DFT diagonals:

$$\mathcal{S}(A, d) = \sum_{c \in \{0, \ldots, C-1\}} \sum_{\substack{(i,j) \in \{0, \ldots, N-1\} \\ i+j=d}} |\mathcal{F}_{i,j}(A_c)|, \tag{3}$$

We additionally denote the first $n$ entries of $\mathcal{S}(A)$ as

$$\mathcal{S}_n(A) = (\mathcal{S}(A, 1), \ldots, \mathcal{S}(A, n)). \tag{4}$$

We omit the term $S(A, 0)$ corresponding to a constant signal, as changes to this component can be fully captured by the bias of the output layer of a neural network. As such, $\mathcal{S}(A, 0)$ is easy to model and is not affected by changes to the embedding.

The term *spectrum* is sometimes used to describe the 2-dimensional DFT of an image. However, to the best of our knowledge, it has not been widely applied in the context of INRs, and no prior work has introduced the spectrum as a vector.

**Wasserstein distance** is a distance function between probability distributions. For distributions P and Q over $\mathbb{R}$, and a cost function $c(x, y) = |x - y|$, it is defined as

$$\mathcal{W}(P, Q) = \inf_{\pi \in \Gamma(P, Q)} \int_{\mathbb{R} \times \mathbb{R}} c(x, y) \, d\pi(x, y), \tag{5}$$

where $\Gamma(P, Q)$ is the set of all joint distributions on $\mathbb{R} \times \mathbb{R}$ with marginals $P$ and $Q$. Although complex to estimate in multiple dimensions, in our 1-dimensional, discrete case the Wasserstein distance is the L1 norm of the difference between cumulative distribution functions: $||\text{CDF}(Q) - \text{CDF}(P)||_1$ (Panaretos & Zemel, 2019).

Table 3: Average PSNR on image representation tasks. FreSh outperforms or matches the baseline performance without introducing significant computational costs (see Table 4). Results for *Wire* are provided for reference only, as it is incompatible with FreSh (see section 4.1). The best results in each section are bolded. Results are averaged over 3 seeds.

| PSNR ↑ | Average | Chest X-Ray | FFHQ-1024 | FFHQ-wild | Kodak | Wiki Art |
|---|---|---|---|---|---|---|
| *Siren* | $33.85 \pm 0.01$ | $37.35 \pm >0.01$ | $37.54 \pm 0.04$ | $34.32 \pm 0.01$ | $31.60 \pm 0.03$ | $28.45 \pm 0.01$ |
| +FreSh | $\mathbf{34.62} \pm 0.01$ | $\mathbf{37.99} \pm 0.01$ | $\mathbf{39.11} \pm 0.01$ | $\mathbf{35.40} \pm 0.01$ | $\mathbf{31.78} \pm 0.02$ | $\mathbf{28.80} \pm 0.01$ |
| *Fourier* | $32.12 \pm 0.01$ | $36.96 \pm 0.03$ | $35.01 \pm 0.04$ | $32.65 \pm 0.01$ | $28.84 \pm 0.04$ | $27.15 \pm 0.01$ |
| +FreSh | $\mathbf{33.45} \pm 0.02$ | $\mathbf{37.77} \pm 0.04$ | $\mathbf{36.81} \pm 0.06$ | $\mathbf{34.62} \pm 0.01$ | $\mathbf{30.06} \pm 0.01$ | $\mathbf{28.01} \pm 0.02$ |
| *Finer* | $\mathbf{35.11} \pm >0.01$ | $\mathbf{38.63} \pm 0.02$ | $\mathbf{40.45} \pm 0.01$ | $\mathbf{36.48} \pm 0.02$ | $\mathbf{31.40} \pm 0.02$ | $\mathbf{28.57} \pm 0.01$ |
| +FreSh | $35.03 \pm 0.01$ | $38.51 \pm 0.04$ | $40.31 \pm 0.07$ | $\mathbf{36.48} \pm 0.01$ | $31.31 \pm 0.03$ | $28.54 \pm 0.01$ |
| *Finer$_{k=0}$* | $34.81 \pm 0.01$ | $\mathbf{38.44} \pm 0.01$ | $39.91 \pm 0.02$ | $35.96 \pm 0.01$ | $\mathbf{31.43} \pm 0.02$ | $28.31 \pm 0.01$ |
| +FreSh | $\mathbf{34.88} \pm 0.03$ | $38.43 \pm 0.03$ | $\mathbf{40.12} \pm 0.06$ | $\mathbf{36.28} \pm 0.02$ | $31.16 \pm 0.01$ | $\mathbf{28.44} \pm 0.05$ |
| *Wire* | $33.54 \pm 0.02$ | $37.96 \pm 0.02$ | $38.13 \pm 0.04$ | $35.04 \pm 0.01$ | $28.95 \pm 0.06$ | $27.62 \pm 0.01$ |

## 4.2 FreSh

FreSh performs a parameter sweep in which the Discrete Fourier Transform and the Wasserstein distance are used to approximate model performance, instead of the costly model training required for grid search.

**Method description.** Our goal is to select an embedding configuration $\theta_i$ from a set of $M$ configurations $\{\theta_i\}_{i \in \{1, \ldots, M\}}$ that would maximize performance when fitting a target image $Y \in \mathbb{R}^{C \times N \times N}$. We propose to use the configuration $\theta_i$ where the associated output of the model at initialization, $\hat{Y}_{\text{init}}^i$, has a similar frequency distribution to the target image, in other words $\mathcal{S}_n(Y) \approx \mathcal{S}_n(\hat{Y}_{\text{init}}^i)$.

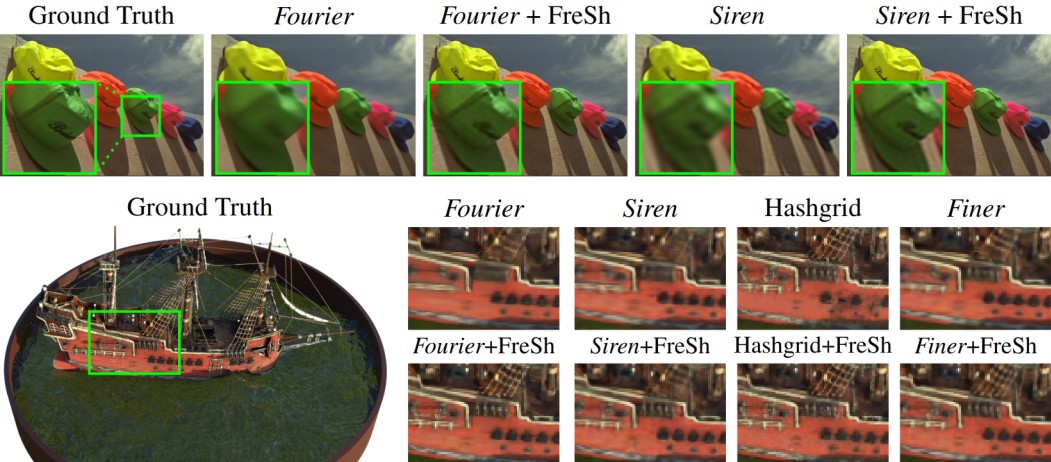

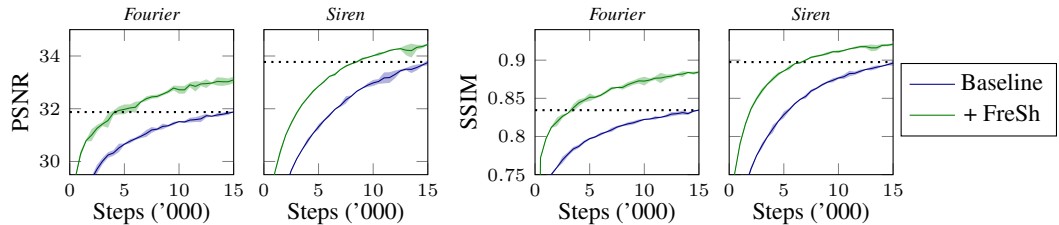

Figure 3: Example model outputs for image modeling (top) and NeRF (bottom). FreSh representations are better at modeling high-frequency details such as text or ropes. For additional examples, see appendix E.

Figure 4: Mean PSNR and SSIM values during training on 50 images (averaged over 3 seeds). FreSh improves the final performance and speeds up convergence. Dotted lines indicate the final results of the baseline model. Shaded area is the 99% confidence interval.

We note that the spectrum $\mathcal{S}_n(A)$ is absolutely homogeneous ($\mathcal{S}_n(\alpha A) = |\alpha|\mathcal{S}_n(A)$), which implies that scaling does not affect the relative presence of different frequencies. Additionally, it is equivalent to scaling the original signal, a common data pre-processing step. As such, the spectrum can be interpreted as a probability distribution, making the Wasserstein distance a natural choice for a similarity measure. This requires only that we use the normalized spectrum, defined as $\tilde{\mathcal{S}}_n(A) = \frac{\mathcal{S}_n(A)}{\|\mathcal{S}_n(A)\|_1}$. With this, we define the FreSh configuration $\theta_j$ as the one that minimizes the Wasserstein distance between the target signal and the model, meaning

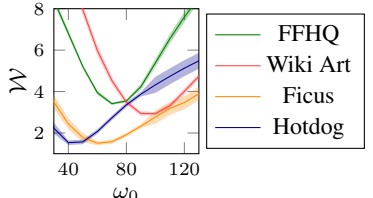

$$j = \arg\min_i \mathcal{W}(\tilde{\mathcal{S}}_n(Y), \tilde{\mathcal{S}}_n(\hat{Y}_{\text{init}}^i)) \qquad (6)$$

$$= \arg\min_i ||\text{CDF}(\tilde{\mathcal{S}}_n(Y)) - \text{CDF}(\tilde{\mathcal{S}}_n(\hat{Y}_{\text{init}}^i))||_1. \qquad (7)$$

Figure 5: Wasserstein distance for selected image and NeRF datasets across different *Siren* configurations. It follows a smooth trend with a distinct global minimum, indicating stable and predictable behavior. Shaded area represents the 95% confidence interval.

In settings where multiple target images are available (video approximation and NeRF), we select one at random to calculate the Wasserstein distance. We visualize the entire selection process for images in Figure 2, and provide a full algorithm of FreSh in the Appendix (see algorithm 1).

**Measurement noise.** Due to the randomness of the image used as the target signal, $Y$, on video approximation and NeRF tasks and the randomness of the model output $\hat{Y}_{\text{init}}^i$ arising from random network weights, the measurement of the Wasserstein distance is noisy. To prevent this from affecting the selection process, we measure the Wasserstein distance 10 times and use its mean to select

the optimal configuration. This is particularly important on video approximation and NeRF tasks as there both $Y$ and $\hat{Y}_{\text{init}}^i$ are sources of noise. This results in higher measurement variance in those tasks (see Figure 5). The FreSh process could be further optimized to use less compute by decreasing the number of measurements, especially on low-noise image representation tasks, but we do not investigate it in detail as FreSh already requires only a negligible amount of compute.

**Spectrum size.** The reason for using a cropped spectrum (equation 4), instead of the full spectrum (equation 3), is due to the noise present in real-world signals. Synthetic signals such as $\hat{Y}_{\text{init}}$ have predominately low-frequency components (see Figure 2), which makes them inherently mismatched with real-world signals in the high-frequency range. As such, using equation 3 (or increasing the spectrum size $n$) would shift FreSh embeddings towards higher frequencies. On the other hand, low values of $n$ can remove high-frequency (not noise) components of the target signal, leading to worse performance. We found setting $n$ to $64$ achieves good results, with additional improvements possible through adjusting $n$, which we further investigate in an ablation study in the Appendix. Although our method introduces a new hyperparameter, it is not sensitive to the target signal, making it easy to configure. Additionally, the spectrum size could be used as an implicit regularizer, as lowering it leads to lower frequencies being selected.

## 5 EXPERIMENTS

FreSh increases results quality and learning speed by tailoring the embedding to the target signal, which improves modeling across all frequencies (see Figure 6) without enormous computational costs of conventional grid searches (see Table 4). We show performance improvements on signal representation tasks and on an inverse problem in the form of estimating radiance fields (Mildenhall et al., 2021). In all experiments, we use FreSh to select one embedding configuration from $\sigma \in \{1, 2, \ldots, 20\}$, $\omega_0 \in \{10, 20, \ldots, 200\}$, $\omega \in \{10, 20, \ldots, 200\}$ and $k \in \{0.0, 0.1, \ldots, 3.0\}$ (parameters are described in table 2). Unless stated otherwise, we set the spectrum size hyperparameter, $n$, to $64$. All experiments are implemented in PyTorch (Paszke et al., 2019) and use the Adam optimizer (Kingma & Ba, 2014).

Table 4: Total time (h) needed to train all models (*Siren*, *Fourier*, *Finer*, *Finer*$_{k=0}$) for the image fitting task with grid search and FreSh, assuming that 20 hyperparameter values are tested (except *Finer* with 31 values). Time of grid search was estimated based on training time of a single configuration. FreSh is an order of magnitude faster. Measurements are averaged over 3 seeds.

| Time (h) $\downarrow$ | Baseline | Grid Search | FreSh |
|---|---|---|---|
| *Siren* | 29.1 $\pm_{0.3}$ | 584.2 $\pm_{6.6}$ | 30.3 $\pm_{0.4}$ |
| *Fourier* | 23.5 $\pm_{0.5}$ | 470.2 $\pm_{9.4}$ | 24.5 $\pm_{0.4}$ |
| *Finer* | 33.0 $\pm_{0.1}$ | 1022.7$\pm_{2.2}$ | 36.0 $\pm_{0.3}$ |
| *Finer*$_{k=0}$ | 31.5 $\pm_{0.1}$ | 629.6 $\pm_{2.5}$ | 34.6 $\pm_{0.3}$ |

Calculating both the Fourier transform and Wasserstein distance is computationally cheap relative to the cost of training. This is a crucial advantage of our method over a trial and error approach, as it makes it feasible to test multiple embedding configurations. We measured the highest relative cost of running FreSh on a high-resolution FFHQ-wild image, where the time required by FreSh reached 50 seconds per tested configuration, which is equivalent to 1% of the training time.

### 5.1 SIGNAL REPRESENTATION

We evaluate on image and video overfitting using the first 10 images from FFHQ (Karras et al., 2019) (both the "in the wild" and cropped images at a resolution of 1024x1024), Wiki Art (Saleh & Elgammal, 2015), Chest X-Ray (Kermany et al., 2018), and Kodak (Franzen, 2024) datasets. For videos, we use the "bikes" and "cat" videos from (Sitzmann et al., 2020b). Image heights and widths for the image fitting task range from 380 to 6720. In video overfitting, we use the current state-of-the-art solution in the form of ResFields (Mihajlovic et al., 2024), which improves the capacity of the MLP by making the weights time-dependent.

We report the average PSNR across all image datasets after full training (15k steps) in Table 3, example outputs in Figure 3 and training curves in Figure 4. FreSh consistently achieves results that are either better than or comparable to the respective baselines, achieving similar performance about 2 times faster in the case of *Siren* and about 4 times faster in the case of Fourier features, while being an order of magnitude less costly to execute than conventional grid search (see Table 4).

Modifying the hyperparameters of *Finer* did not lead to any noticeable improvements. Nonetheless, FreSh performs comparably well, being especially helpful when the bias is eliminated ($k = 0$). In this situation, FreSh successfully reproduces the performance of the original model with bias, raising questions about the need for both parameters. The advantage of FreSh is especially visible in terms of perceived quality, as measured by SSIM. These improvements are achieved through better approximation of the target signal across all frequency components (see Figure 6) thanks to the increased frequency magnitudes in the embedding layer. Similarly to optimal configurations from Table 1, configurations selected by FreSh are highly varied, highlighting the suboptimality of any constant configuration (see Appendix E).

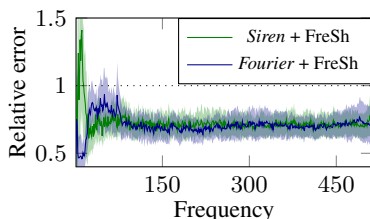

Figure 6: The mean spectrum of residuals for the image representation task expressed as a percentage of the baseline configuration performance, as measured on Kodak images. The dotted line indicates the baseline performance. FreSh improves modeling across all frequencies, lowering the error by about 30%. The shaded area represents the 95% confidence interval.

To show the versatility of FreSh, we successfully apply it to videos, which contain temporal changes not directly measured by our method. We measure PSNR and SSIM and report the results in Table 5, with a full table available in the Appendix (see Table 9). We found that to improve the model's frequency modeling, we needed to remove time from the input coordinates, making it an indirect input only through the weight modifications of ResFields. This suggests that commonly used embedding strategies are ineffective for signals with qualitatively different directions (see Figure 16).

## 5.2 Neural Radiance Fields

Neural radiance fields (Mildenhall et al., 2021) are used for synthesizing novel views of a 3D object based on a limited number of measurements (images) of the object. Unlike in the image fitting task, the target signal is unknown and spectra calculations required by FreSh are performed on the available images. To render views $\hat{Y}_{\text{init}}$ from the model, we take a single sample along each ray in the middle of the scene, where we assume the volume density is maximal. We perform experiments using a torch implementation (Tang, 2022; Tang et al., 2022) of InstantNGP (Müller et al., 2022) and synthetic NeRF data. Due to the complexity of this task, we adjusted the spectrum size for each model, and use 128 for Hashgrid, 64 for *Fourier* and *Finer*, and 32 for the spectrally-concise *Siren*.

FreSh achieves similar or better reconstruction quality than baseline configurations and positional encoding, with the exception of axis-aligned datasets (lego, materials) where positional encodings have an advantage due to their frequencies also being axis-aligned (see Table 6). In the case of the Hashgrid embedding, we found that the recommended range for the resolution parameter $N_{\max} \in [512, 524288]$ induced frequencies that are too high, with FreSh selecting resolutions outside this range ($N_{\max} \in \{64, 128, 256\}$). This lowering of embedding frequencies leads to higher reconstruction quality, highlighting the applicability of FreSh in various settings and detecting models with both too low and too high frequencies.

Table 5: Mean PSNR on video representation. FreSh outperforms baseline embedding configurations and the NeRF-like embedding (Positional Encoding). Each configuration was tested with and without time as an input coordinate. The model benefits from embeddings reconfigured with FreSh only when time is not an input, indicating that different frequency magnitudes are required for spatial and temporal directions. Results for Positional Encoding are provided for reference only, as it is incompatible with FreSh (see section 4.1). Results are averaged over 3 seeds.

|  | Time Input | Cat PSNR ↑ | Bikes PSNR ↑ |
|---|---|---|---|
| *Siren* | ✓ | 38.72 ±0.05 | 41.12 ±0.03 |
| +FreSh | ✓ | 36.84 ±0.03 | 40.28 ±0.01 |
| *Siren* | ✗ | 39.84 ±0.03 | 40.39 ±0.02 |
| +FreSh | ✗ | **40.61**±0.04 | **41.62**±0.01 |
| *Fourier* | ✓ | 38.14 ±0.07 | 40.92 ±0.07 |
| +FreSh | ✓ | 37.38 ±0.04 | 40.29 ±0.03 |
| *Fourier* | ✗ | 38.82 ±0.03 | 40.61 ±0.03 |
| +FreSh | ✗ | **39.68**±0.04 | **41.13**±0.02 |
| Pos. enc. | ✓ | **37.39**±0.01 | **39.78**±0.04 |
| Pos. enc. | ✗ | 37.13 ±0.04 | 39.57 ±0.05 |
| Hashgrid | ✓ | 34.71 ±0.15 | 37.13 ±>0.01 |

Table 6: Average PSNR achieved with Positional Encoding (also known as NeRF), *Siren*, *Fourier*, *Finer* and Hashgrid embeddings on synthetic NeRF datasets. FreSh improves performance in many instances. Results for positional encoding are provided for reference only, as it is incompatible with FreSh (see section 4.1).

| PSNR ↑ | Average | Chair | Drums | Ficus | Hotdog | Lego | Materials | Mic | Ship |
|---|---|---|---|---|---|---|---|---|---|
| *Siren* | 31.21±0.09 | 34.20±0.11 | 26.03±0.13 | 30.25±0.06 | **35.81**±0.13 | 30.28±0.14 | **29.67**±0.11 | 34.25±0.08 | 28.60±0.18 |
| +FreSh | **31.43**±0.09 | **34.71**±0.11 | **26.15**±0.14 | **31.03**±0.06 | 35.70±0.13 | **30.81**±0.15 | 29.40±0.12 | **34.41**±0.07 | **28.74**±0.19 |
| *Fourier* | 32.04±0.09 | 34.70±0.13 | 26.42±0.12 | 31.19±0.05 | 35.97±0.14 | 31.58±0.15 | **31.67**±0.10 | 34.38±0.09 | 30.07±0.12 |
| +FreSh | **32.63**±0.09 | **36.32**±0.11 | **26.92**±0.14 | **32.38**±0.05 | **36.26**±0.14 | **33.41**±0.17 | 30.99±0.12 | **34.43**±0.09 | **30.58**±0.19 |
| *Finer* | **31.64**±0.09 | 34.64±0.11 | **26.12**±0.13 | **31.31**±0.06 | 35.75±0.14 | 31.59±0.13 | **29.71**±0.11 | **34.35**±0.08 | **29.55**±0.14 |
| +FreSh | 31.62±0.09 | **34.86**±0.11 | 26.10±0.13 | 31.13±0.06 | **35.77**±0.14 | **31.80**±0.13 | 29.51±0.11 | 34.27±0.07 | 29.47±0.17 |
| $Finer_{k=0}$ | 31.39±0.09 | 34.38±0.11 | 26.02±0.13 | 30.80±0.06 | **35.72**±0.14 | 31.08±0.13 | **29.68**±0.11 | 34.18±0.08 | 28.94±0.17 |
| +FreSh | **31.62**±0.09 | **34.84**±0.11 | 26.02±0.13 | **31.23**±0.06 | 35.66±0.14 | **31.66**±0.13 | 29.47±0.11 | **34.37**±0.08 | **29.60**±0.13 |
| Hashgrid | 31.09±0.09 | **34.04**±0.13 | 25.77±0.09 | 30.65±0.05 | 34.82±0.18 | 31.42±0.13 | 28.67±0.09 | **34.34**±0.08 | **29.06**±0.18 |
| +FreSh | **31.22**±0.09 | 33.55±0.14 | **26.17**±0.11 | **31.05**±0.05 | **35.22**±0.14 | **32.56**±0.14 | **28.92**±0.09 | 34.09±0.07 | 28.71±0.20 |
| Pos. enc. | 32.22±0.09 | 35.46±0.11 | 26.04±0.12 | 30.77±0.06 | 35.75±0.14 | 33.77±0.16 | 31.78±0.11 | 35.15±0.08 | 29.72±0.10 |

## 6 LIMITATIONS

While this work represents an initial step toward automating embedding configuration for INRs, FreSh has several limitations inherited from existing embedding methods. Notably, FreSh does not account for direction-dependent frequency magnitudes, potentially impacting performance in scenarios where directionality is important, such as video approximation. Although the spectrum size hyperparameter is easier to configure than traditional embedding parameters due to its independence from the target signal, it still can require manual tuning. Moreover, FreSh performance is inherently limited by the constraints of the embeddings it configures and cannot achieve quality better than a conventional grid search. Specific to FreSh, it is not applicable to models that utilize excessively high embedding frequencies, such as NeRF or *Wire*.

## 7 CONCLUSION

Hyperparameter selection is crucial for INR performance, yet there is limited research in this area. This makes evaluating new architectures costly due to the need for extensive grid searches, or unfair when suboptimal hyperparameter values are used. We address these challenges by introducing FreSh, a model-agnostic method for configuring coordinate embeddings that significantly reduces the cost of finding effective configurations compared to parameter sweeps. FreSh leverages frequency information to select the configuration that best aligns with the target signal, effectively biasing the model to fit all relevant frequencies. While FreSh is not compatible with certain models, such as *Wire*, it proves highly effective when applicable, facilitating the use of improved embeddings like Fourier features. This is particularly relevant in the context of Neural Radiance Fields (NeRF), where the adoption of this embeddings has been limited by high hyperparameter sensitivity. Although FreSh introduces a new hyperparameter, it is not sensitive to the target signal and it requires little to none adjustment. By utilizing ResFields, we have observed that frequency magnitudes in certain tasks are significantly direction-dependent. This suggests that new embeddings may be needed to account for this dependence, and expanding FreSh to also consider directional dependencies is a promising research direction that could further enhance its effectiveness.

ACKNOWLEDGMENTS

The research of A. Kania was funded by the program Excellence Initiative – Research University at the Jagiellonian University in Kraków. The work of P. Spurek was supported by the National Centre of Science (Poland) Grant No. 2023/50/E/ST6/00068.

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

## A  OPTIMIZATION OF EMBEDDING WEIGHTS

The embedding layer of an INR requires careful tuning of its hyperparameters (as discussed in section 3), even though its parameters should be optimized by gradient descent. To explain this phenomenon, we investigate the embedding of *Siren* and observe a pattern similar to that noted by Tancik et al. (2020) about the *Fourier* model: gradient descent fails to effectively optimize the embedding layer. This is likely caused by the periodic nature of activation function used in the embedding.

**Measuring frequency of an embedding neuron**
Let's recall that the *Siren* embedding is given as

$$\text{embedding}(\mathbf{x}) = \sin(\omega_0 \mathbf{W} \mathbf{x} + \mathbf{b}), \qquad (8)$$

where $\mathbf{W}$ are the embedding weights. Denoting the $i$-th row of $\mathbf{W}$ as $\mathbf{w}_i$, we analyze the frequency of each embedding neuron independently, ignoring the bias term $\mathbf{b}$, as it only affects phase, not frequency magnitude. Specifically, we consider the frequency of $\sin(\omega_0 \mathbf{w}_i \cdot \mathbf{x})$, which is maximal along the direction specified by $\mathbf{w}_i$. Given that $\mathbf{w}_i \cdot \mathbf{x} = ||\mathbf{w}_i||_2 ||\mathbf{x}||_2 \cos(\angle(\mathbf{w}_i, \mathbf{x}))$, we can simplify the analysis to the frequency of $\sin(\omega_0 ||\mathbf{w}_i||_2 ||\mathbf{x}||_2 \cos(\angle(\mathbf{w}_i, \mathbf{x}))) = \sin(\omega_0 ||\mathbf{w}_i||_2 ||\mathbf{x}||_2)$.

Frequency is defined as the number of full periods a function completes over a unit distance. By choosing $2\pi$ as the unit distance, the frequency of $\sin(||\mathbf{x}||_2)$ is simply 1. The frequency of $\sin(\omega_0 ||\mathbf{w}_i||_2 ||\mathbf{x}||_2)$ scales proportionally and is given by $\omega_0 ||\mathbf{w}_i||_2$.

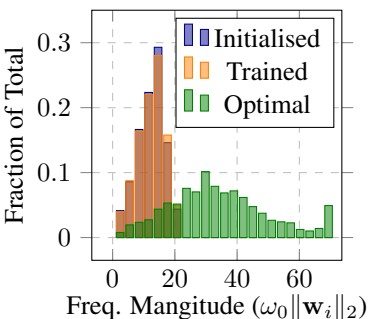

Histogram of Embedding Frequencies

Figure 7: Frequency magnitudes of the embedding layer of *Siren* for baseline configurations (at initialization and after training) and FreSh configuration (at initialization) during training on 5 images from the experiment in section 3. During training, magnitudes of the baseline model fail to increase to a scale comparable with the optimal configurations, highlighting that embedding frequencies are not effectively optimized by gradient descent. As frequency magnitudes remain largely constant during training, they must be configured as a hyperparameter. Magnitudes were clipped to range $[0, 80]$.

**Experiment** We investigate how frequency magnitudes change during training on 5 images (we use the same images as in section 3) by comparing the distributions of frequency magnitudes between models at initialization and after training (see Figure 7). We additionally provide frequency magnitudes of embeddings configured using grid search (see Table 1) as a reference of optimal magnitudes. We use the same experimental setup as in section 5, only changing whether the embedding weights are optimized or not.

During training the magnitudes change only slightly, but this increase is negligible when compared to magnitudes induced by optimal embeddings, whose size reflects the multiple-fold increase of $\omega_0$ observed in Table 1. Since gradient descent fails to notably increase embedding frequencies, the magnitudes must be adjusted as a hyperparameter, significantly increasing the cost of finding optimal models.

## B  ARCHITECTURES FROM RELATED WORKS

This section covers architectures that were left out of the main text.

**NeRF** (Mildenhall et al., 2021) maps 5D coordinates - spatial location (x, y, z) and viewing direction ($\theta, \varphi$) - to volume density and view-dependent emitted radiance, which are then used to render novel views of a scene. It employs positional embedding (Vaswani et al., 2017), which is a multiresolution sequence of $L$ frequencies:

$$\gamma_P(\mathbf{x}) = [\sin(2^0 \mathbf{x}), \cos(2^1 \mathbf{x}), \dots, \sin(2^{L-2} \mathbf{x}), \cos(2^{L-1} \mathbf{x})]. \qquad (9)$$

$\gamma_P$ helps overcome the spectral bias of MLPs but is biased toward axis-aligned directions, which can result in performance loss depending on the rotation of the target object (Tancik et al., 2020). This embedding works well with frequencies which are higher than the main components of the target

signal (e.g., $L = 16$). When training NeRF models, we embed the spatial coordinates using $\gamma_P$ and the viewing direction with the spherical harmonics basis (Fridovich-Keil et al., 2022; Verbin et al., 2022), following the approach of Müller et al. (2022).

**Fourier features** (Tancik et al., 2020) is a direction-invariant alternative to eq. (9) which densely samples Fourier basis functions. Although such sampling is not feasible in realistic settings, it can be well approximated through random sampling (Rahimi & Recht, 2007), resulting in the mapping

$$\gamma_F(\mathbf{x}) = [\sin(2\pi\mathbf{W}\mathbf{x}), \cos(2\pi\mathbf{W}\mathbf{x})], \tag{10}$$

where weights are sampled from an isotropic frequency distribution, such as Gaussian $\mathbf{W} \sim \mathcal{N}(0, \sigma)$. Scale of this distribution, $\sigma$, controls frequency magnitudes and it was found to be an important factor in the final performance of a model (Tancik et al., 2020). Even though the performance of Fourier features was shown to be better than that of eq. (9), it has not been widely adopted, possibly due to the high sensitivity of this embedding to the value of $\sigma$. We refer to this embedding as *Fourier*.

***Finer*** (Liu et al., 2024) is a variation of the *Siren* model with a broader supported frequency set. It extends the frequency set of *Siren* by utilizing a variable-periodic activation, $\varphi(x) = \sin((|x|+1)x)$, in its embedding

$$\gamma_F(\mathbf{x}) = \varphi(\omega(\mathbf{W}\mathbf{x} + \mathbf{b})), \tag{11}$$

where $\mathbf{W} \in \mathbb{R}^{m \times d}$ is the matrix of weights and $b \in \mathbb{R}^m$ is bias. This model controls frequencies through the scaling parameter $\omega$ and the width of bias distribution $k$ ($\mathbf{b} \sim \text{Uniform}(-k, k)$) [1]. The authors of *Finer* suggest using bias as the main method for increasing the model capacity to approximate high-frequency signals. However, a similar effect can also be achieved by adjusting $\omega$, which raises questions about the necessity of using bias. We consider two scenarios: one where $\omega$ is fixed at 30 and $k$ is optimized, and another where bias is removed ($k = 0$) and $\omega$ is optimized. We find that bias can be removed without significantly affecting performance (see Table 3). We denote the model with no bias as *Finer*$_{k=0}$.

**Multiresolution hash encoding** (Hashgrid) divides the space into increasingly finer grids (e.g., 16), assigning each voxel to a random feature vector via a hash table (Müller et al., 2022). The authors recommend adjusting specific hyperparameters for each dataset, particularly the size of the hash table, which impacts the memory footprint, and the resolution of the finest level, $N_{\max}$, which affects the size of details that can be easily modelled. As a baseline, we use a resolution of $2048$. At low resolutions, the number of grid vertices is of comparable size to the number of hash table entries, effectively limiting the total parameter count of low resolution models. To prevent different parameter counts from affecting the performance, we set a relatively low hash table size of 8192 entries.

***Wire*** uses a continuous complex Gabor wavelet activation function

$$\psi(x) = e^{j\omega_0 x} e^{-|s_0 x|^2}, \tag{12}$$

where $\omega_0$ controls frequency and $s_0$ controls width of the wavelet, with typically used values of $\omega_0 = 20$ and $s_0 = 10$. This activation makes model design problematic, as it increases frequencies at every layer of the model, making frequencies of the model depth-dependent. Moreover, we note that, similarly to NeRF, *Wire* uses extremely high frequencies, which are incompatible with FreSh. As such, we report results of *Wire* only for context.

## C  ADDITIONAL SPECTRUM EXAMPLES

In this section we consider untrained models, showing their outputs and comparing their spectra to the Kodak "bikes" image (see Figure 2). As illustrated in Figure 8, the Wasserstein distance effectively highlights that using high-frequency embedding configurations would be beneficial for *Siren* and *Fourier*, as their spectra are closer to the spectrum of the signal. We note that the feature sizes in most model outputs align with those typically found in natural images. However, models such as *Wire* produce extremely small, high-frequency features, implying that they perform well despite using high frequencies, possibly by taking advantage of some additional mechanism.

---

[1] Saragadam et al. (2023) denote $\omega$ as $\omega_0$. We removed the index to make hyperparameters of *Finer* and *Siren* easier to differentiate.

## D  PSEUDOCODE FOR FRESH

We present the pseudocode for FreSh in Algorithm 1. The algorithm is slightly different between image and video/NeRF tasks, due to multiple images being available for the latter. This is reflected in the definition of $Y_{sample}$ which consist of multiple images - one for each measurement of the Wasserstein distance. It is constructed by sampling 10 images for NeRF and video approximation tasks, while for the image approximation task, $Y_{sample}$ consists of the same image repeated 10 times. Even though the target signal is not random for the image approximation task, it is also measured multiple times due to the randomness of the model output.

---

**Algorithm 1:** FreSh

---

**Input:** $\Theta$ set of embedding configurations, n > 0
**Data:** $Y_{sample}$ a sample of images, $X$ input coordinates
**Output:** $\theta_{\text{best}}$ embedding configuration with the lowest Wasserstein distance
$d_{\text{best}} \leftarrow \infty$;
$\theta_{\text{best}} \leftarrow$ None;
**for** $\theta \in \Theta$ **do**
  $distances \leftarrow []$;
  **for** $Y$ $in$ $Y_{sample}$ **do**
    $S_{\text{target}} \leftarrow$ full_spectrum$(Y)[: n]$;
    $\phi \leftarrow$ get_random_model_weights$()$;
    $S_{\text{model}} \leftarrow$ full_spectrum$(\text{model}(\theta, \phi, X))[: n]$;
    $d \leftarrow$ wasserstein_distance$(S_{\text{model}}, S_{\text{target}})$;
    $distances$.append$(d)$;
  $d_{\text{mean}} \leftarrow$ mean$(distances)$;
  **if** $d_{mean} < d_{best}$ **then**
    $d_{\text{best}} \leftarrow d_{\text{mean}}$;
    $\theta_{\text{best}} \leftarrow \theta$;

**return** $\theta_{\text{best}}$;

---

## E  ADDITIONAL SIGNAL REPRESENTATION AND RECONSTRUCTION RESULTS

Table 7: **Ablation study of spectrum size** on image approximation task. All runs in the table were conducted with a fixed seed. The optimal spectrum size depends primarily on the base architecture, with the dataset having minimal impact. The best results in each section are bolded.

| | $n$ | Average | Chest X-Ray | FFHQ-1024 | FFHQ-wild | Kodak | Wiki Art |
|---|---|---|---|---|---|---|---|
| *Siren* + FreSh | 32 | 34.28 | 37.49 | 38.13 | 35.00 | **32.13** | 28.67 |
| *Siren* + FreSh | 64 | **34.59** | 38.00 | **39.09** | **35.37** | 31.72 | **28.78** |
| *Siren* + FreSh | 128 | 34.15 | **38.37** | 38.66 | 34.93 | 30.41 | 28.37 |
| *Fourier* + FreSh | 32 | 33.00 | 37.33 | 36.03 | 34.06 | 29.93 | 27.68 |
| *Fourier* + FreSh | 64 | 33.47 | 37.80 | 36.83 | 34.64 | **30.04** | **28.05** |
| *Fourier* + FreSh | 128 | **33.55** | **38.10** | **37.13** | **34.89** | 29.58 | 28.03 |
| *Finer* + FreSh | 32 | 34.80 | 38.43 | 39.87 | 35.96 | **31.44** | 28.30 |
| *Finer* + FreSh | 64 | **35.06** | 38.59 | **40.44** | **36.50** | 31.23 | **28.56** |
| *Finer* + FreSh | 128 | 34.97 | **38.76** | 40.09 | 36.38 | 31.17 | 28.46 |
| *Finer$_{k=0}$* + FreSh | 32 | 34.44 | 38.07 | 39.16 | 35.33 | **31.42** | 28.19 |
| *Finer$_{k=0}$* + FreSh | 64 | **34.95** | 38.50 | **40.27** | 36.32 | 31.13 | **28.53** |
| *Finer$_{k=0}$* + FreSh | 128 | 34.79 | **38.68** | 40.10 | **36.35** | 30.61 | 28.24 |

**Spectrum size**  Spectrum size, $n$, controls the number of frequencies included in the spectrum (eq. (4)). The size of the spectrum has to be limited due to the inherent mismatch between natural and synthetic signals (as discussed in section 4). Given that the architecture primarily determines

the characteristics of the spectrum (e.g., *Siren* was designed to have a concise spectrum), the specific target signal is not expected to play a significant role in selecting the optimal spectrum size. Our ablation study in Table 7 supports this, showing that the source of the dataset has minimal impact on the optimal spectrum size. Since the optimal spectrum size does not depend on the target signal, fine-tuning this value is significantly simplified, as it can be selected using a small subset of target signals. After determining the optimal spectrum size for a given architecture, no further tuning should be needed.

In general, we find that FreSh performs well when the spectrum size $n$ is set to $64$, although performance can be further improved through fine-tuning (see Table 7). Interestingly, *Siren*-based models gain no benefit from increasing the spectrum size beyond $64$. This is likely because *Siren* is designed with hidden layers that have minimal impact on frequencies (Sitzmann et al., 2020b), leading to a concise spectrum, which is well described even with a shorter spectrum. In contrast, the *Fourier* model benefits from larger spectra, likely because no special considerations were made in its design to keep the spectrum concise.

**Selected hyperparameter values**    We report configurations selected by FreSh on the image approximation task in Figure 9. Similarly to the optimal configurations (see Table 1), we observe high variability in model configurations selected by FreSh. Even when narrowed to a single dataset (e.g. Kodak), the configurations are varied, highlighting the need for adjusting the embedding frequencies for each target signal.

**Decreasing the default embedding frequency**    In almost all experiments in section 5, models benefit from embedding configurations that induce higher frequencies than the baseline configuration. However, the Hashgrid model is an exception, where FreSh improves performance by selecting configurations that induce lower frequencies. In this section, we present another such example by testing *Siren* on a low-frequency synthetic dataset.

We generate the target signal as a sum of sinusoids with up to 5 periods and train *Siren* using the same setting as described in section 5. Since the target signal is relatively simple, we reduce the training time by a factor of 10. We found that with the default learning rate training is not stable and PSNR can decrease by us much as 40% in 100 steps, which is why we additionally lower the learning rate by a factor of 10. For this dataset, FreSh selects configuration of $\omega_0 = 10$, resulting in frequencies three times lower than those in the baseline model. This low-frequency embedding improves the results, as shown in Figure 10. This shows that FreSh improves quality by aligning frequencies and not by simply increasing them.

**Learning speed**    Re-configuring a model with FreSh increases the learning speed, with image approximations being sharp after as little as 1000 iterations. We include example outputs from all image datasets in Figures 11 to 14.

**Video approximation**    We report PSNR and SSIM scores on the video approximation task in Table 9. Similarly to PSNR, SSIM is highest with models configured using FreSh and without a time input, making it an indirect input only through the weight modifications of ResFields. This could be explained by different characteristics of the video signal along temporal and spatial directions (see Figure 16).

**Object reconstruction**    We provide additional examples in Figure 15.

**INR Improvement Methods**    Recent advances have introduced methods to mitigate the spectral bias in INRs, such as Batch Normalization (Cai et al. (2024)) and Fourier Reparameterization (Shi et al. (2024)). Meanwhile, Saratchandran et al. (2024) propose the *From Activation to Initialization* (FAI) method, which optimizes initialization based on the activation function but does not directly address spectral bias. We evaluate FreSh alongside most of these approaches to highlight their complementary nature (see Table 8). We used the same experimental setup as in section 5. We don't report results using Batch Normalization, as using it yielded results worse than baseline *Siren*.

Table 8: **FreSh compared with other INR improvement methods.** Other than FreSh, we evaluate improved weight initialization (*From Activation to Initialization* or FAI) and Fourier Reparameterization (FR). FreSh improves the performance of other methods.

| PSNR ↑ | Average | Chest X-Ray | FFHQ-1024 | FFHQ-wild | Kodak | Wiki Art |
|---|---|---|---|---|---|---|
| *Siren* | 33.85 ±0.01 | 37.35 ±>0.01 | 37.54 ±0.04 | 34.32 ±0.01 | 31.60 ±0.03 | 28.45 ±0.01 |
| +FreSh | 34.62 ±0.01 | 37.99 ±0.01 | 39.11 ±0.01 | 35.40 ±0.01 | 31.78 ±0.02 | **28.80**±0.01 |
| + FAI | 34.32 ±0.01 | 37.86 ±0.01 | 38.33 ±0.01 | 34.84 ±0.01 | **31.97**±0.03 | 28.61 ±0.01 |
| + FAI + FreSh | **34.87**±0.01 | **38.33**±0.01 | **39.66**±0.01 | **35.77**±0.01 | 31.79 ±0.03 | **28.80**±0.01 |
| + FR | 33.79 ±0.01 | 37.50 ±>0.01 | 37.51 ±0.04 | 33.98 ±0.01 | **31.65**±0.03 | 28.32 ±0.02 |
| + FR + FreSh | **34.28**±>0.01 | **37.80**±0.01 | **38.66**±0.01 | **34.82**±0.02 | 31.54 ±0.03 | **28.57**±0.01 |

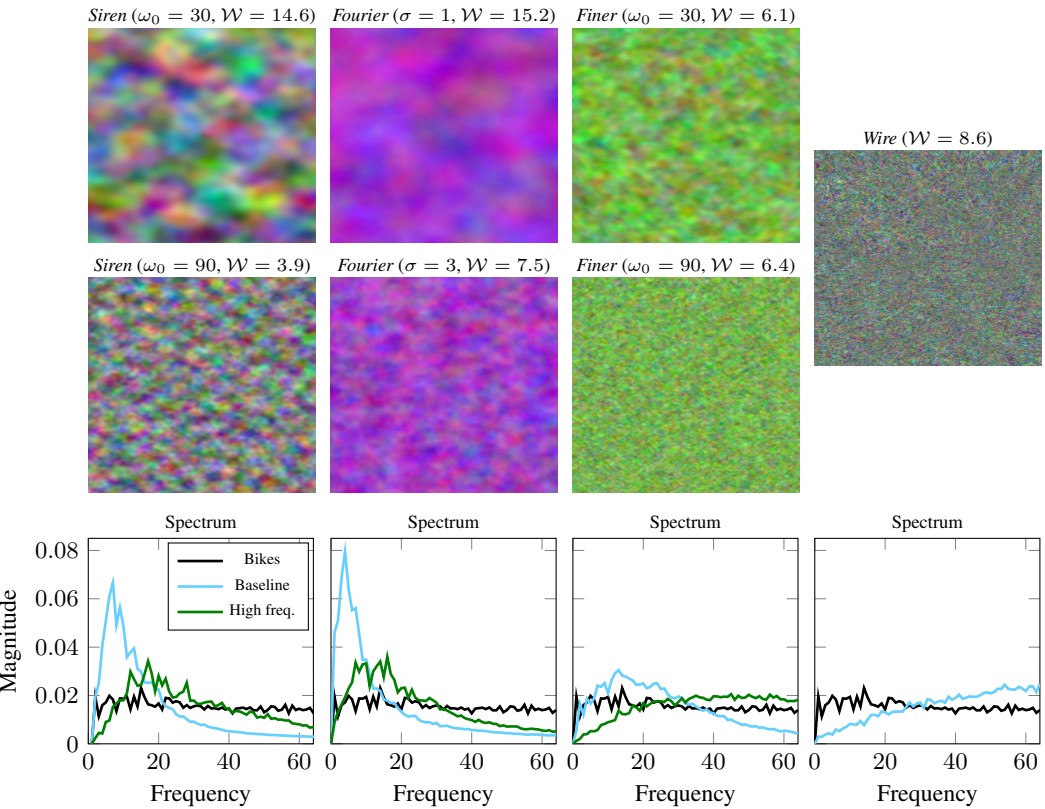

Figure 8: Example outputs from untrained models (top two rows) and their corresponding spectra (bottom row). For comparison, the spectrum of the Kodak "bikes" image is also provided, along with the Wasserstein distances between the model configurations and the target image. We consider a baseline (top row) and a high-frequency (middle row) configuration for each model other than *Wire*, where the baseline is already high-frequency. The Wasserstein distance effectively shows that *Siren* and *Fourier* baseline configurations are too low-frequency for the highly detailed "bikes" image. Notably, the uniformly colored regions in outputs from models other than *Wire* (and the high-frequency *Finer*) tend to be comparable in size or larger than typical features in natural images. In contrast, the output from *Wire* exhibits extremely fine, high-frequency features.

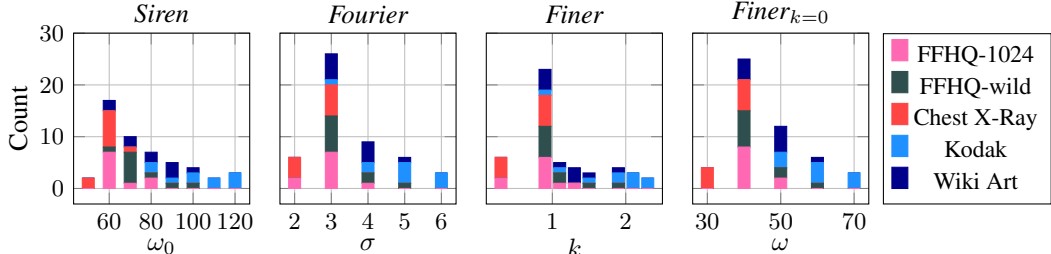

Figure 9: Embedding configurations selected by FreSh for the image representation task. Hyperparameter values are highly varied even within datasets, highlighting the need for fine-tuning the embedding layer for each target signal.

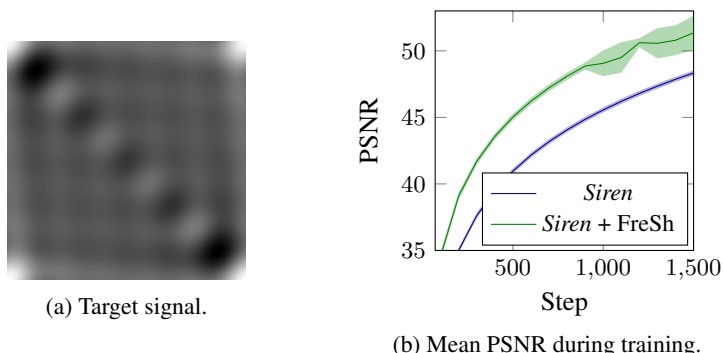

(a) Target signal.

(b) Mean PSNR during training.

Figure 10: Training of *Siren* on an low frequency image. (a) Target image created as a sum of sinusoids of at most 5 periods. (b) PSNR values averaged over 10 seeds. *Siren* using the default configuration ($\omega_0 = 30$) is slower in fitting the signal than a low-frequency *Siren* configured with FreSh ($\omega_0 = 10$). This example demonstrates that reducing the model frequency can sometimes be advantageous, and FreSh is capable of identifying such situations.

Table 9: Video representation results for NeRF, *Siren* and *Fourier* embeddings. FreSh outperforms baseline embedding configurations and NeRF. Each configuration was tested with and without time as an input coordinate. The model benefits from embeddings reconfigured with increased frequencies only when time is not an input, indicating that different frequency magnitudes are required for spatial and temporal directions. Results are averaged over 3 seeds.

| | Use | Cat | | Bikes | |
| | time | PSNR | SSIM | PSNR | SSIM |
|---|---|---|---|---|---|
| *Siren* | ✓ | $38.72 _{\pm0.05}$ | $0.9518 _{\pm0.0004}$ | $41.12 _{\pm0.03}$ | $0.9677 _{\pm0.0001}$ |
| +FreSh | ✓ | $36.84 _{\pm0.03}$ | $0.9444 _{\pm0.0002}$ | $40.28 _{\pm0.01}$ | $0.9684 _{\pm0.0002}$ |
| *Siren* | ✗ | $39.84 _{\pm0.03}$ | $0.9553 _{\pm0.0001}$ | $40.39 _{\pm0.02}$ | $0.9657 _{\pm0.0001}$ |
| +FreSh | ✗ | $\mathbf{40.61}_{\pm0.04}$ | $\mathbf{0.9579}_{\pm0.0002}$ | $\mathbf{41.62}_{\pm0.01}$ | $\mathbf{0.9708}_{\pm0.0001}$ |
| *Fourier* | ✓ | $38.14 _{\pm0.07}$ | $0.9481 _{\pm0.0006}$ | $40.92 _{\pm0.07}$ | $0.9678 _{\pm0.0006}$ |
| +FreSh | ✓ | $37.38 _{\pm0.04}$ | $0.9483 _{\pm0.0002}$ | $40.29 _{\pm0.03}$ | $0.9696 _{\pm>0.0001}$ |
| *Fourier* | ✗ | $38.82 _{\pm0.03}$ | $0.9514 _{\pm0.0003}$ | $40.61 _{\pm0.03}$ | $0.9667 _{\pm0.0001}$ |
| +FreSh | ✗ | $\mathbf{39.68}_{\pm0.04}$ | $\mathbf{0.9555}_{\pm0.0002}$ | $\mathbf{41.13}_{\pm0.02}$ | $\mathbf{0.9700}_{\pm0.0002}$ |
| NeRF | ✓ | $\mathbf{37.39}_{\pm0.01}$ | $\mathbf{0.9471}_{\pm0.0002}$ | $\mathbf{39.78}_{\pm0.04}$ | $\mathbf{0.9675}_{\pm0.0002}$ |
| NeRF | ✗ | $37.13 _{\pm0.04}$ | $0.9462 _{\pm0.0002}$ | $39.57 _{\pm0.05}$ | $0.9664 _{\pm0.0004}$ |

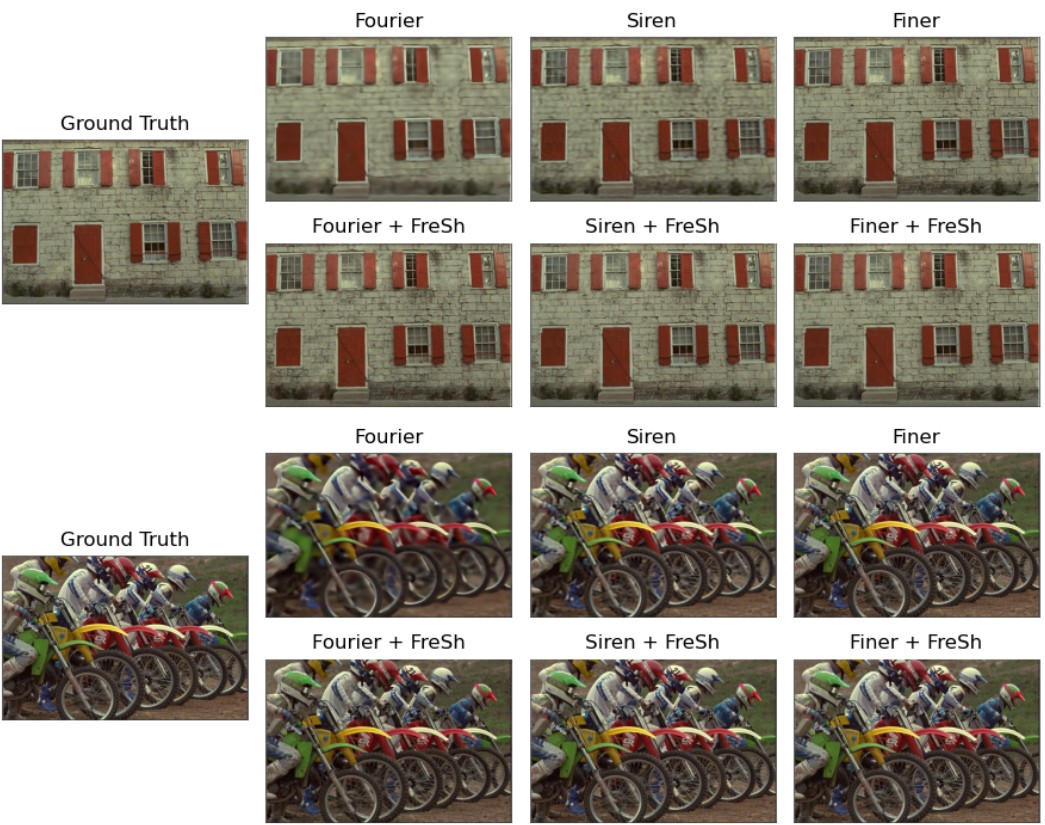

Figure 11: Model comparison on Kodak images after 1000 iterations of training.

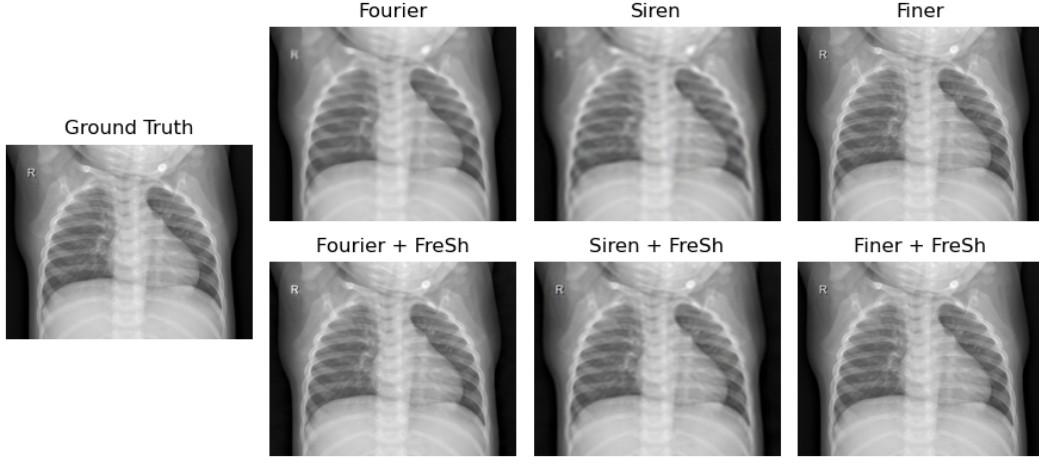

Figure 12: Model comparison on a Chest X-Ray image after 1000 iterations of training.

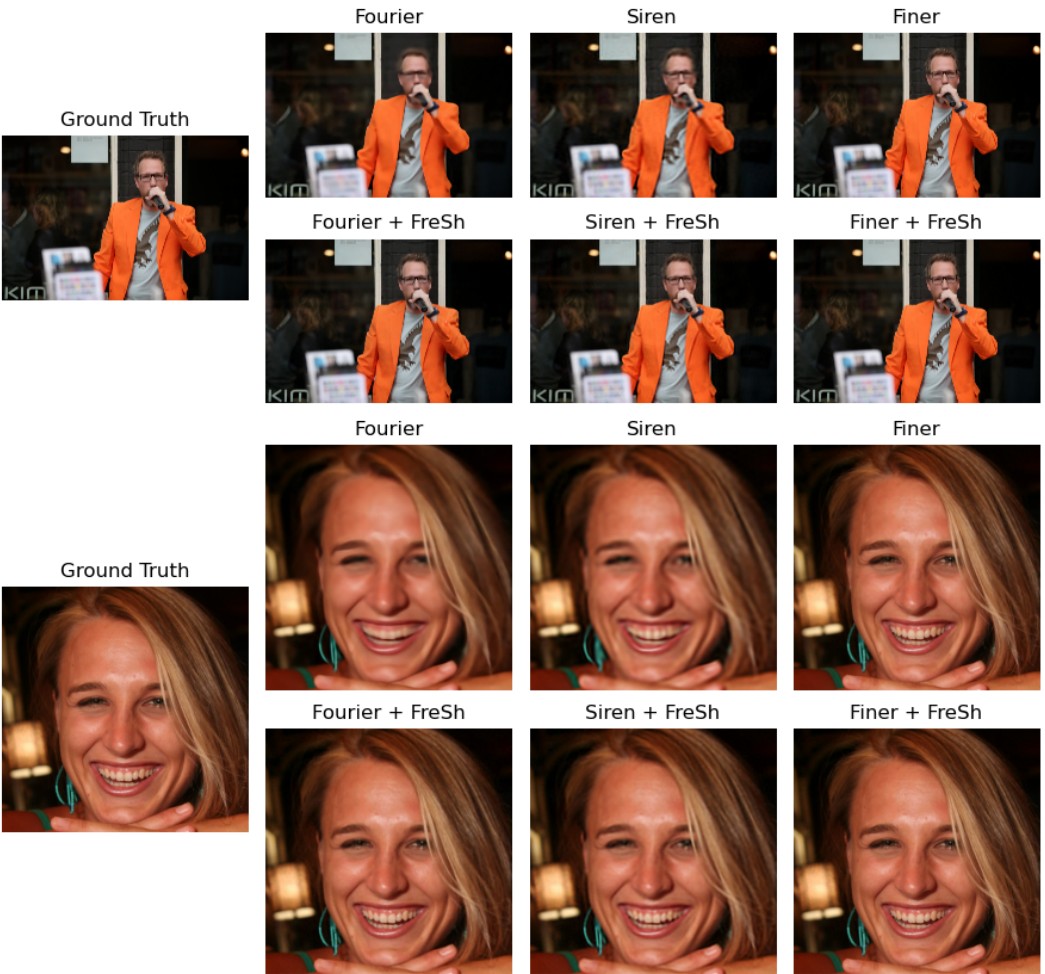

Figure 13: Model comparison on images from FFHQ-wild and FFHQ-1024 after 1000 iterations of training.

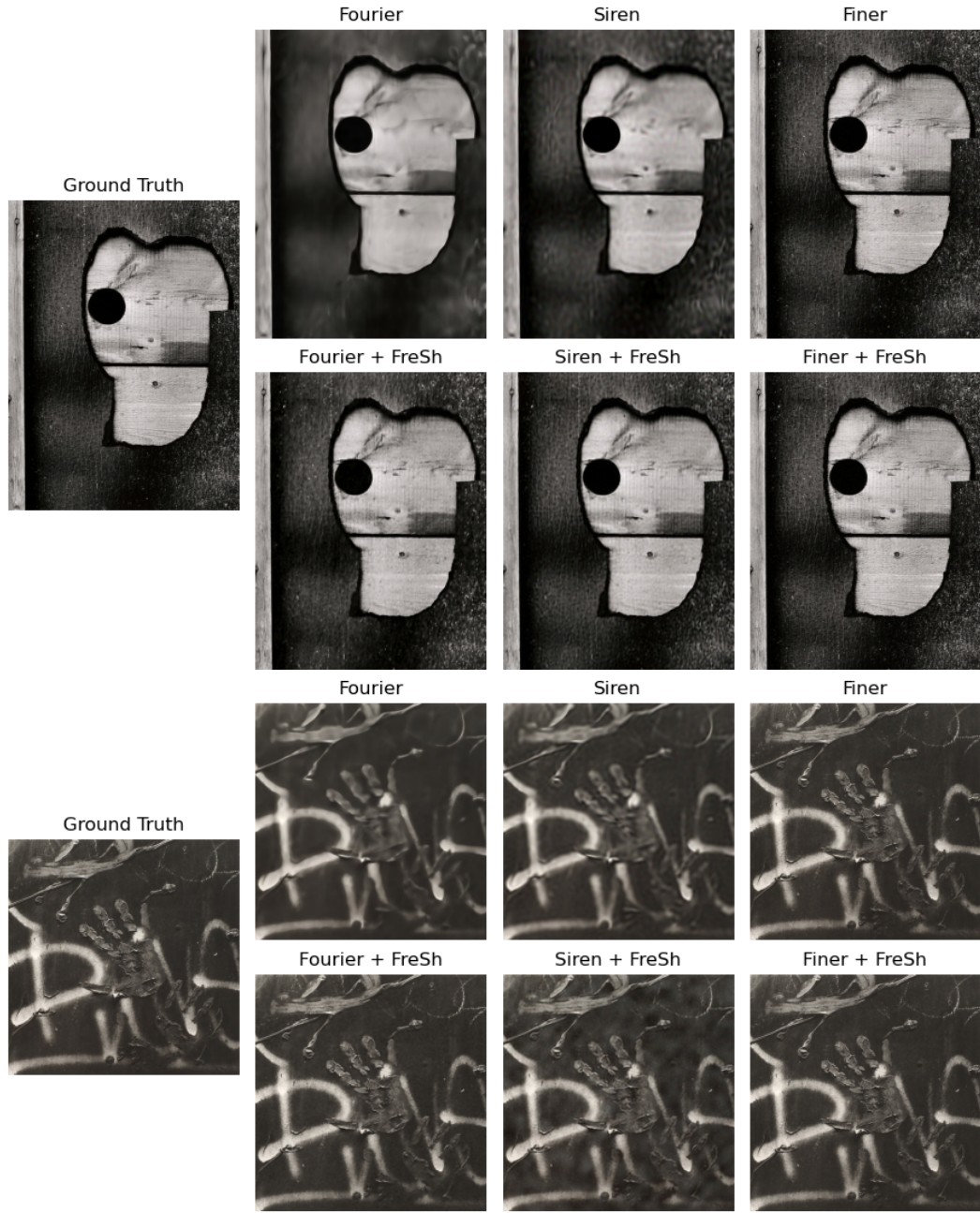

Figure 14: Model comparison on Wiki Art images after 1000 iterations of training.

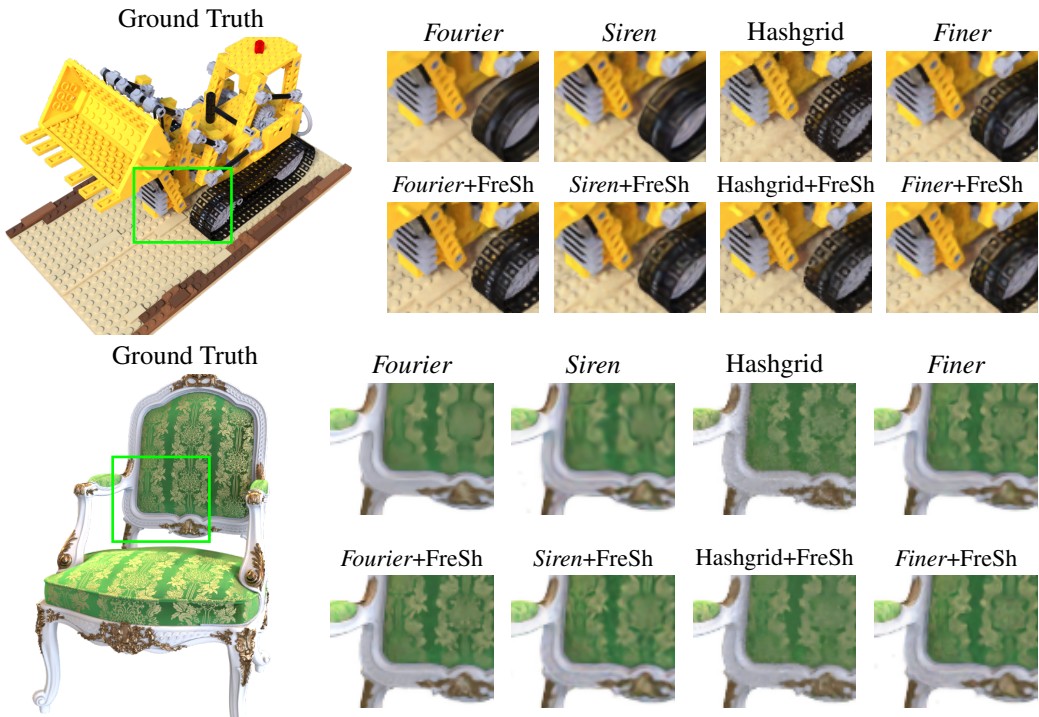

Figure 15: Example model outputs for the object modeling task after approximately 20% of the training.

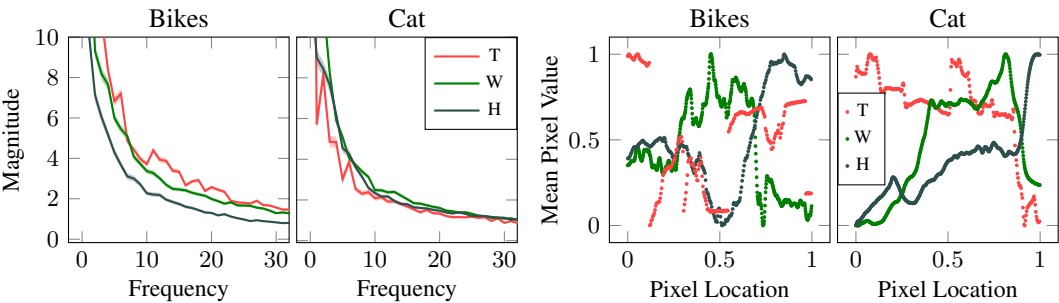

Figure 16: **Axis-aligned analysis of video signals.** We compute the average DFT of the video signal over 1000 randomly selected axis-aligned vectors and measure the mean pixel values along three directions: Time (T), Width (W), and Height (H). Signal along the time direction rapidly changes, such as during scene transitions, leading to spiky frequency distributions. This qualitative difference between temporal and spatial coordinates suggests that they should be treated differently in an embedding layer. Pixel locations and values were normalized to the range $[0, 1]$ to account for variations in video sizes and pixel value ranges.

