# OpenReview forum: "FreSh: Frequency Shifting for Accelerated Neural Representation Learning"
_ICLR.cc/2025/Conference — ICLR 2025 Poster_

### Official Review · Reviewer_TDK4 · 2024-10-28

**Soundness:** 3
**Presentation:** 4
**Contribution:** 2
**Rating:** 6
**Confidence:** 4

**Summary:**

The authors notice that the initial frequency spectrum of an untrained model’s output correlates strongly with the model’s eventual performance on a given target signal. They want to “pretrain” the untrained model output with the end signal through a hyperparameter search over initialization parameters.

**Strengths:**

The paper is well written, and introduces a novel method of initialization by fitting the frequency spectrum of the untrained model with the desired downstream task. The logic is sound, and this concept may apply to other fields.

**Weaknesses:**

Evidence
- The experimental evidence is extremely weak in my opinion and does not demonstrate a significant increase that would warrant other researchers’ usage of this method. Most results show improvements of less than 1dB, which is essentially identical to the naked eye.
- The results in Table 1 are concerning. After 15,000 steps the difference between the Base Siren model and using Fresh have virtually no difference (PSNRs of 31.18 and 31.81).
- Table 3 - Why does FreSH have no improvement on Finer and Finer with k=0 and most samples? If it is only applicable to the SIREN and Fourier features model it would be way too limited to warrant an accept as these are several years old baselines.
- Moreover in Table 6, overall the results are not impressive as most PSNRs increase by < 1dB (i.e. Finer has virtually no change)
- Table 8, why does Fresh hurt the results when you include time? It seems as expected input for video.

Baselines
- Paper seems to have a lot of overlap with SPDER https://arxiv.org/pdf/2306.15242 (especially in theory, and that the untrained model output aligns with end reconstruction) and should compare the baseline against it.
- Other baselines should be (Ramasinghe & Lucey, 2022) which suggests Gaussian activations. It is mentioned in the paper but not reported on to the best of my knowledge.
- Instant NGP (HashGrid?) may also be used in the image baseline, as it excels in high-resolution images but is not included here for that.
- Overall, SIREN seems to be a very weak baseline model. I understand that tuning the omega_0 parameter can improve results, but I don’t believe this to be as significant of a contribution on top of the existing model.

Complexity
- Although computationally cheap (utilizing random initialization and DFT), it adds much more complexity to the setup
- For example, on the Wasserstein measurement setup, “To prevent this from affecting the selection process, we measure the Wasserstein distance 10 times and use its mean to select.” I feel this is too much of a hack to be reliably used by other researchers, but I may yield to what other reviewers say.
- Authors could clarify what the parameter values represent in the first section of Experiments for ease of reading.

Impact
- It is stated the method is incompatible for “extremely high frequencies” (i.e. NeRF and Wire), which makes me doubt its generalizability.

**Questions:**

- Why not just have these parameters be learned rather than grid searched (as I believe omega_0 can be learnable at least, I am not sure about the others)?
- 15,000 steps seems very excessive to fit a single image, how much wall clock does this take on a gpu for good results? Other methods like instant-ngp and spder (above) are much faster.
- In 5.2 in experiments why is NeRF not used as a baseline also as it is mentioned in the paper? I am not sure which one it is in Table 6.

---

> ### Author Response · Authors · 2024-11-22
> **Evidence**
>
> Thank you for your feedback. We address the particular comments below.
>
> > The experimental evidence is extremely weak in my opinion and does not demonstrate a significant increase that would warrant other researchers’ usage of this method. Most results show improvements of less than 1dB, which is essentially identical to the naked eye.
>
> We agree with the reviewer that the **absolute differences are small**. This is due to our decision to **train until convergence** (addressed in the next paragraph). To demonstrate that these **differences are significant**, we repeat experiments three times and report standard errors. For instance, in the image experiments, the standard error is very small (approximately 0.03), meaning the **absolute differences are large relative to the standard error**.
>
> > The results in Table 1 are concerning. After 15,000 steps the difference between the Base Siren model and using Fresh have virtually no difference (PSNRs of 31.18 and 31.81).
>
> The **differences are significant due to the small standard errors**. We chose long training times to **allow models to converge, ensuring a fair comparison**. Shorter training times would artificially inflate differences, but could lead to misleading results.
>
> Our training length aligns with standard practices in the field; for example, Siren (Sitzmann et al., 2020) uses 10,000 steps for image experiments. Similarly, works like WIRE (Saragadam et al., 2023), Finer (Liu et al., 2024), and Fourier Features (Tancik et al., 2020) train INRs for several thousand steps.
>
> > Table 3 - Why does FreSH have no improvement on Finer and Finer with k=0 and most samples? If it is only applicable to the SIREN and Fourier features model it would be way too limited to warrant an accept as these are several years old baselines.
>
> While FreSh does not improve Finer in the image setting, Finer with k=0 shows improvements for most datasets. Additionally, FreSh achieves **state-of-the-art results on video** (table 5) and provides multiple improvements over baselines in the NeRF task (Siren, Fourier, Finer with k=0, Hashgrid). Furthermore, Siren and Fourier features remain **important baselines whose performance is often underreported due to suboptimal embeding configurations** - an issue FreSh directly addresses.
>
>
> > Moreover in Table 6, overall the results are not impressive as most PSNRs increase by < 1dB (i.e. Finer has virtually no change)
>
> The results are significant due to the **small standard errors**. The absolute differences appear small because **we train models to convergence**, ensuring a fair comparison.
>
> > Table 8, why does Fresh hurt the results when you include time? It seems as expected input for video.
>
> Time is always incorporated via the weight modifications in ResFields. However, we observe a performance decrease when it is also provided as an input to the embedding layer. Please see our general comment for a detailed discussion of this phenomenon.

---

> ### Author Response · Authors · 2024-11-22
> **Baselines**
>
> > Paper seems to have a lot of overlap with SPDER https://arxiv.org/pdf/2306.15242 (especially in theory, and that the untrained model output aligns with end reconstruction) and should compare the baseline against it.
>
> Thank you for highlighting the SPDER paper, which, like our work, emphasizes the importance of spectral bias. However, SPDER’s approach is distinct from ours. **SPDER addresses spectral bias by designing an architecture** optimized for learning various frequency components, similar to Siren (Sitzmann et al., 2020), WIRE (Saragadam et al., 2023), and Finer (Liu et al., 2024). In contrast, **FreSh enhances existing architectures** by configuring their frequency-controlling hyperparameters, making it complementary to architectural innovations like SPDER.
>
>
> Given SPDER's promising results, we attempted to test it as a baseline. However, we identified limitations in its applicability, particularly to high-resolution images (see experiment below) and tasks like NeRF (see SPDER paper).
>
> Experiment: Re-implementing SPDER in the FreSh codebase
>
> We replicated a simplified version of Table 3 from our paper, testing on one image per dataset under the same training settings. We tested SPDER, Siren ($\omega_0=30$) and Siren + FreSh:
>
> |               | Chest X-Ray | FFHQ-1024 | FFHQ-wild | Kodak    | Wiki Art |
> |---------------|-------------|-----------|-----------|----------|----------|
> | SPDER         | 37.1        | 33.3      | 32.5      | 21.1*    | 21       |
> | Siren         | 38.3        | 35.1      | 33.7      | **26.8** | 22.2     |
> | Siren + Fresh (ours) | **38.8**    | **36.5**  | **34.6**  | 26.4     | **22.7** |
> (*PSNR during training was higher, at 25.1, but the model collapsed.)
>
> **In our setting, SPDER underperformed**, which we don’t believe warrants using it as a baseline for our study. We speculate the lower performance of SPDER can be attributed to the following:
>
> **PSNR calculation bug in SPDER**
> SPDER calculates PSNR as
>
> $\text{PSNR}=20 \log_{10}(\frac {\text{MAX}_I} {\text{MSE}})$
>
> instead of the correct
>
> $\text{PSNR}=20 \log_{10}(\frac {\text{MAX}_I} {\sqrt{\text{MSE}}})$
>
> See [SPDER GitHub](https://github.com/katop1234/SPDER/blob/763aaed0ada2d0761d40e229aaca712f985d0854/helpers.py#L210) for the relevant code.
> $\text{MAX}_I$ is the maximum possible pixel value.
>
> **Overparameterization dependency of SPDER**
>  SPDER’s original experiments involved low-resolution images and large networks (5 hidden layers). In contrast, our setup with high-resolution images and smaller networks (3 hidden layers) led to a performance degradation, suggesting that SPDER relies on relatively high parameter counts to perform well.
>
> **Evaluation protocol**
> SPDER evaluated on the entire image, whereas we tested on unseen pixels.
>
>
>
>
> > Other baselines should be (Ramasinghe & Lucey, 2022) which suggests Gaussian activations. It is mentioned in the paper but not reported on to the best of my knowledge.
>
> We report the results of WIRE (Saragadam et al., 2023), which is a direct generalization of Gaussian activations and has been shown to perform better.
>
> > Instant NGP (HashGrid?) may also be used in the image baseline, as it excels in high-resolution images but is not included here for that.
>
> Thank you for the suggestion. Our experiments are set up in different repositories, so we were unable to quickly add InstantNGP as an image baseline. However, we have updated the paper to include InstantNGP for the video experiment (see table below). We observe that InstantNGP does not generalize well to unseen pixels, which aligns with findings from Mihajlovic et al., 2023.
> |               | Cat   | Bikes |
> |---------------|-------|-------|
> | InstantNGP    | 34.71 | 37.13 |
> | Siren + Fresh  (ours)| 40.61 | 41.62 |
>
>
> > Overall, SIREN seems to be a very weak baseline model. I understand that tuning the omega_0 parameter can improve results, but I don’t believe this to be as significant of a contribution on top of the existing model.
>
> Although Siren was proposed a few years ago, we believe it remains an important baseline model, **still used in recent works** like ResFields (Mihajlovic et al., 2023) and SPDER, and **achieving state-of-the-art results on video** (Mihajlovic et al., 2023), which we further improve.
>
> Often, when used as a baseline, the $\mathbf{\omega_0}$ **parameter of Siren is not fine-tuned**, leading to suboptimal performance. We consider this an **important issue**, as it complicates fair comparisons with novel architectures. Including Siren in our work aims to encourage researchers to compare against well-tuned Siren models, increasing confidence in new methods.
>
> Furthermore, **we report improvements for models beyond Siren**, such as Fourier Features, Finer, and InstantNGP.

---

> > ### Author Response · Authors · 2024-11-22
> >
> > > Although computationally cheap (utilizing random initialization and DFT), it adds much more complexity to the setup
> > > For example, on the Wasserstein measurement setup, “To prevent this from affecting the selection process, we measure the Wasserstein distance 10 times and use its mean to select.” I feel this is too much of a hack to be reliably used by other researchers, but I may yield to what other reviewers say.
> >
> > We have **updated Figure 2 to clarify the FreSh pipeline** and make the algorithm easier to understand.
> >
> > While our method introduces some added complexity as an extension to existing solutions, we believe this complexity is minimal, especially since we provide an automatic script to run the FreSh pipeline. The **simplicity of our approach has also been noted by other reviewers** (swg8, cowA, vNDz).
> >
> >
> > > Authors could clarify what the parameter values represent in the first section of Experiments for ease of reading.
> >
> > We have added a reference to Table 2 in the relevant section to clarify the meaning of the parameter values for ease of reading.
> >
> > > It is stated the method is incompatible for “extremely high frequencies” (i.e. NeRF and Wire), which makes me doubt its generalizability.
> >
> > **We demonstrate the generalizability of our method by applying it to four different architectures (Siren, Fourier, Finer, Hashgrid) and three distinct tasks**. Notably, FreSh works with Instant NGP, which is based on a data structure rather than a neural network. Additionally, the Fourier + FreSh model outperforms Positional Encoding (NeRF) on the NeRF task, and Siren + FreSh achieves better results than Wire on the image task.
> >
> > For further discussion of high-frequency models/embeddings, please refer to our general response.
> >
> > > Why not just have these parameters be learned rather than grid searched (as I believe omega_0 can be learnable at least, I am not sure about the others)?
> >
> > As we understand the problem, the **loss landscape is too complex for efficient optimization of such parameters**, and in some architectures (e.g., Instant NGP), they cannot be optimized during training.
> >
> > We discuss a similar issue in Section A of the appendix (“Optimization of embedding weights”), where we note that the embedding layer weights do not significantly change their scale during training, which is equivalent to changing $\omega_0$.
> >
> > > 15,000 steps seems very excessive to fit a single image, how much wall clock does this take on a gpu for good results? Other methods like instant-ngp and spder (above) are much faster.
> >
> > It takes around 30 minutes to train a model for 15,000 steps, but this heavily depends on the image resolution. With FreSh, training time can be reduced. For example, Fourier Features trained for 15,000 steps achieve similar results to Fourier Features configured with FreSh trained for 5,000 steps (corresponding to approximately 10 minutes of training).
> >
> > While Instant NGP is faster than neural-network-based approaches, it generalizes worse, as noted by Mihajlovic et al. (2023) and observed in our video experiments. Additionally, we show that FreSh can improve results achieved by Instant NGP. We are unaware of any purely neural-network-based methods that are significantly faster and broadly applicable (see our comments on SPDER and training time). For comparison, we report training times from the SPDER experiment  we reported above:
> >
> >
> > | Time (minutes) | Chest X-Ray  | FFHQ-1024  | FFHQ-wild  | Kodak  | Wiki Art |
> > |----------------|--------------|------------|------------|--------|----------|
> > | SPDER          | 90           | 69         | 85         | 42     | 78       |
> > | Siren          | 54           | 37         | 51         | 20     | 45       |
> > | Siren + FireSh (ours) | 51           | 31         | 37         | 19     | 36       |
> >
> > We observe that training SPDER takes longer than training Siren for the same number of iterations. We attribute this difference to the more complex activation function used by SPDER. If it was optimized, the training time for SPDER would likely be comparable to Siren.
> >
> >
> > > In 5.2 in experiments why is NeRF not used as a baseline also as it is mentioned in the paper?
> >
> > NeRF is indeed used as a baseline, but **we refer to the NeRF model as "positional encoding" (pos. enc.)** throughout the paper. Since we use the term "NeRF" to refer to the 3D object reconstruction task, we avoided using the same name for the model itself. To clarify, we have updated the table description to include the phrase “positional encoding (also known as NeRF)”. Additionally, we’ve revised sections of the paper where "NeRF" was used instead of "positional encoding."

---

> ### Comment · Reviewer_TDK4 · 2024-11-26
> **Reviewer Response**
>
> I thank the authors for their diligence in the responses. I am impressed with the depth of their justifications and am willing to update my score to recommend acceptance.
>
> However, I will note:
> - The PSNR changes are concerningly low, and I am not convinced by the argument that the standard errors in the change being small compared to the mean change justifies the method's use. (In fact, this would imply that one can say with confidence that the benefits are minor). As a suggestion, the authors should add a convincing reason in the abstract/intro as why other researchers should opt to use this (i.e. it will speed up training by x% on average for high resolution images). But just my personal take.

---

> > ### Author Response · Authors · 2024-11-28
> >
> > We sincerely thank the reviewer for considering the revised manuscript and additional experiments and for adjusting the score. Following the reviewers comment, in the final version of the manuscript we will better highlight the advantages of FreSh, such as its potential applicability to future architectures and its computational efficiency, which is approximately 20 times lower than that of grid search (in our setting).

---

### Official Review · Reviewer_swg8 · 2024-10-28

**Soundness:** 2
**Presentation:** 2
**Contribution:** 3
**Rating:** 6
**Confidence:** 4

**Summary:**

This paper proposes a method for selecting the hyperparameters for current MLP-based INR models to improve the suboptimal representation performance of models with default hyperparameters. This method measures the Wasserstein distance between frequency distributions of initial-state models and the target signal and selects the hyperparameters that minimize the Wasserstein distance.  This paper validates this method on several current INR models via 2D image approximation, 2D video fitting and neural radiance fields. Experimental results demonstrate that this method is effective to some extent.

**Strengths:**

1. The method of this paper is fairly easy to understand and flow. This is the first hyperparameter selection method for implicit neural representation models based on the idea of frequency alignment.
2. The computation cost of this method can be ignored.
3. Through relatively extensive experiments and the experiments on "decreasing the default embedding frequency" in the supplementary materials, this paper well demonstrates that this method could adjust part hyperparameters to adjust the “preferred spectrum” of MLPs, achieve better representation of target signals.

**Weaknesses:**

1. The related work of this paper seems overly lengthy and overlooks some of the latest methods for spectral bias. For example, spectral bias can be overcame by special initialization and training dynamics adjustment, such as reparameterization and batch normalization. Moreover, as stated in the paper that Fresh is a simple initialization technique in line 024, Fresh should compare to the previous initialization method like “From Activation to Initialization: Scaling Insights for Optimizing Neural Fields”.

2. There lacks a reliable theoretical or experimental link between the key observation (line 019) and Fresh (line 241，Table 2).  More concretely, there need an explanation that the adjustment of embedding layer hyperparameters could affect the whole model's spectrum. A more quantitative expression would be better.  The illustration in Fig. 1 seems to be vague. Clarification of this point would certainly raise my opinion of the paper.

3. Some experimental results show that Fresh could not provide a better result even worse than baseline models, such as results with time input in Table 5 and in Table 6. Intuitively, Fresh should not obtain worse results. It might show that there are several unclear relationships. Solving the weakness 2. might help this issue.

4. The experiment in Appendix A seems for demonstrating that weights of embedding layer could not be trained sufficiently by gradient descent. However, why is the optimization algorithm SGD? Meanwhile, is the index L2 norm （Eq. 8） used by the paper (720) meaningful? This experiment should be illustrated more clearly.

**Questions:**

Typically, spectral bias means that neural networks tend to fit the high-frequency component of the target signal. Therefore, spectral bias can be reflected as different learning speeds for different frequency components. Although the advanced activation functions such as Sine improve the representation performance, MLPs with Sine still first fits the low-frequency parts as shown in some paper such as “Improved implicit neural representation with Fourier reparameterzied Training”. Inspired by this phenomenon, these methods might not change the bias; rather, it broadens the range of frequencies the model can represent. Under this perspective, Fresh might shift some special spectrum distribution that enjoys the fastest learning speeds of MLPs towards the target signals. So the detailed description of this special spectrum distribution or the validation of the existence of this special spectrum might be an interesting problem. This point will not affect my scoring. If authors could try to find this special spectrum and do more exploration about this problem, I think that it will further deepen our understanding of deep learning.

---

> ### Author Response · Authors · 2024-11-21
>
> Thank you for your feedback. We address the particular comments below.
>
> > (...) spectral bias can be overcame by special initialization and training dynamics adjustment, such as reparameterization and batch normalization. Moreover, as stated in the paper that Fresh is a simple initialization technique in line 024, Fresh should compare to the previous initialization method like “From Activation to Initialization: Scaling Insights for Optimizing Neural Fields”.
> Thank you for referring us to the insightful paper, “From Activation to Initialization: Scaling Insights for Optimizing Neural Fields” (FAI). We have updated the related work section to include this reference.
>
> We note that the initialization method proposed in FAI can be used in combination with FreSh. To verify this, we trained Siren on one image from each dataset used in our paper, using both the Siren initialization and the one proposed in FAI. Our results show that FreSh improves performance similarly to FAI initialization. Importantly, **both methods can be used together, yielding even better results**.
>
> |                            | Chest X-Ray  | FFHQ-1024  | FFHQ-wild  | Kodak  | Wiki Art |
> |----------------------------|--------------|------------|------------|--------|----------|
> | Siren                      | 38.3         | 35.1       | 33.7       | 26.8   | 22.2     |
> | Siren + Fresh (ours)       | 38.8         | 36.4       | 34.6       | 26.4   | **22.7** |
> | Siren + FAI                | 38.8         | 35.7       | 34.0       | **26.9**| 22.3     |
> | Siren + FAI + Fresh (ours) | **39.3**     | **36.8**   | **34.9**   | 26.3   | **22.7**  |
>
> We believe this simplified experiment demonstrates the potential of combining orthogonal INR improvement methods. A full experiment (including all images and multiple seeds) will be included in the paper at a later stage.
>
>
> > There lacks a reliable theoretical or experimental link between the key observation (line 019) and Fresh (line 241，Table 2). More concretely, there need an explanation that the adjustment of embedding layer hyperparameters could affect the whole model's spectrum. A more quantitative expression would be better. The illustration in Fig. 1 seems to be vague. Clarification of this point would certainly raise my opinion of the paper.
>
> We **updated the appendix with examples illustrating how changes to the frequency parameters of INRs affect their spectra** (Figure 8, page 19). Additionally, we performed an experiment where we initialized 5 different Siren models with varying values of $\omega_0$ and computed the average frequency, weighted by the normalized frequency vector. This experiment demonstrates how increasing hyperparameter values increases the average frequency of a model (see table below).
>
> | $\omega_0$             	| 30 | 60 | 90 | 120  | 150 |
> |----------------------------|----|----|----|------|-----|
> | Weighted mean of frequency | 16 | 24 | 29 | 33   | 35  |
>
> Unfortunately, we find that explaining the frequency spectrum in detail would require more space than is available in Section 1 / Fig. 1. Instead, **we provide a visual explanation of the spectrum in Figure 2 (which we revised)** and have added a reference in the caption of that figure to additional spectrum examples (Figure 8, page 19).
>
>
> > Some experimental results show that Fresh could not provide a better result even worse than baseline models, such as results with time input in Table 5 and in Table 6. Intuitively, Fresh should not obtain worse results. It might show that there are several unclear relationships. Solving the weakness 2. might help this issue.
>
> Regarding the video experiments (Table 5), FreSh improves results when time is not used as an input. We suspect that due to the different characteristics of video signals along the time direction, it requires different treatment than spatial dimensions. Please refer to our discussion of the time input in our general comment.
>
> In the NeRF experiments (Table 6), FreSh does indeed sometimes result in worse performance than the baseline. As this is the first work aiming to perform frequency matching, our method is not yet perfect and occasionally selects suboptimal configurations. We plan to prepare a follow-up work to further analyze the relationship between frequency and performance, particularly in high-frequency models like Wire and NeRF, with the goal of improving FreSh based on these new insights.

---

> ### Author Response · Authors · 2024-11-21
>
> > The experiment in Appendix A seems for demonstrating that weights of embedding layer could not be trained sufficiently by gradient descent. However, why is the optimization algorithm SGD? Meanwhile, is the index L2 norm （Eq. 8） used by the paper (720) meaningful? This experiment should be illustrated more clearly.
>
> Thank you for pointing that out. In the experiment in Appendix A, we used the FreSh codebase. As such, we actually **used Adam as the optimizer**, not SGD. The reference to SGD in the text was intended as a general term for variants of the original SGD algorithm. We clarified this in the revised version of the paper.
>
> We have **updated the text to better clarify the experimental setup** and provide a more detailed explanation of the experiment, including how the frequency is measured. The **L2 norm is used as a measure of vector length** in this context.
>
> > (...) Fresh might shift some special spectrum distribution that enjoys the fastest learning speeds of MLPs towards the target signals. So the detailed description of this special spectrum distribution or the validation of the existence of this special spectrum might be an interesting problem. This point will not affect my scoring. If authors could try to find this special spectrum and do more exploration about this problem, I think that it will further deepen our understanding of deep learning.
>
> FreSh operates under the assumption that the best spectrum for efficient learning is the one most similar to the spectrum of the target signal. Based on our results, this appears to be a reasonable heuristic. However, we acknowledge that it may not be the only mechanism influencing model performance, especially in the case of high-frequency models like Wire. Given the complexity of this topic, we hope to explore this further and provide a more comprehensive explanation in a follow-up work.

---

### Official Review · Reviewer_cowA · 2024-10-28

**Soundness:** 4
**Presentation:** 4
**Contribution:** 3
**Rating:** 8
**Confidence:** 4

**Summary:**

Hyperparameter optimization is a tricky but important problem for implicit neural representations (INRs), since different hyperparameters may be optimal for different signals, but hyperparameter sweeps can be expensive. The authors introduce a fast proxy for determining if an INR’s hyperparameters are well-suited to a particular training signal. Instead of directly optimizing the post-training loss, the authors propose to minimize the Wasserstein distance between frequency distributions of the training signal and the newly initialized INR’s output. This skips the step of actually training each INR, dramatically reducing the time needed to perform a hyperparameter sweep. This proxy appears to perform quite well, conferring most of the performance advantage of a traditional hyperparameter grid search. The authors evaluate their method across a range of image datasets, videos and NeRFs.

**Strengths:**

Clear, simple, effective solution to a practical problem. Thorough demonstrations that it works. The paper is well-written and well-presented, easy to read and understand. I can imagine this technique being widely adopted for INR hyperparameter optimization.

**Weaknesses:**

Section 5 contains a few typos:

Section 5, paragraph 2, “a trail and error approach”

Section 5.1: You fell victim to LaTeX’s backwards quotation marks, one of the classic blunders

Otherwise the paper is quite solid.

**Questions:**

> Similarly, Wire (Saragadam et al., 2023) uses very high frequencies and increases frequencies at each hidden layer, not just at the embedding layer. This makes it incompatible with FreSh

I am curious if you can say more about these models (NeRF and Wire) which use very high frequencies. What purpose do these high frequency components serve if they are not reflected in the training data? Why don’t these models perform poorly as FreSh would predict?

You mention that your FreSh measurements are noisy due to random initialization, so you average across 10 initializations. [1] find that SIREN’s random initialization of the first layer has a significant effect on the final PSNR (Appendix A.6). I would be interested to see if your method could be further improved by using FreSh to select not only the hyperparameters, but also the random initialization to start from.

In the appendix you say that there is high variability in the model configurations selected by FreSh, even within a single dataset. But how much of this variability is random vs. actually important for performance? I would be curious to see how per-image FreSh compares to a dataset-wide FreSh: how much of the advantage of FreSh comes from optimizing for each individual image, and how much can be achieved just by making an appropriate hyperparameter choice for each dataset.

[1] https://openreview.net/pdf?id=iKPC7N85Pf

---

> ### Author Response · Authors · 2024-11-21
>
> Thank you for your feedback and listing all our typos (they are fixed). We address the particular comments below.
>
> > I am curious if you can say more about these models (NeRF and Wire) which use very high frequencies. What purpose do these high frequency components serve if they are not reflected in the training data? Why don’t these models perform poorly as FreSh would predict?
>
> Our hypothesis is that these models (NeRF and Wire) perform well despite not having a “reasonable” spectrum due to mechanisms other than frequency alignment. As far as we know, it’s not well explained in the literature. As they don’t depend on frequency alignment, FreSh cannot predict the performance of these models accurately.
>
> For further discussion on high-frequency models, please refer to our general comment.
>
> > You mention that your FreSh measurements are noisy due to random initialization, so you average across 10 initializations. [1] find that SIREN’s random initialization of the first layer has a significant effect on the final PSNR (Appendix A.6). I would be interested to see if your method could be further improved by using FreSh to select not only the hyperparameters, but also the random initialization to start from.
>
> We find that the noise in the Wasserstein distance measurement primarily arises from the random sampling of images from the target signal in video and NeRF experiments, rather than from weight randomness. Additionally, we observe that the variation in final PSNR is negligibly small in our setting (with 3 hidden layers and 256 hidden neurons). If we were to use networks with higher variability in final PSNR (such as deeper and narrower MLPs, as noted in the paper cited by the reviewer), the impact of random initialization would be more pronounced. In such a setting, it could be possible to choose the model initialisation with FreSh, e.g. by choosing the one with the lowest Wasserstein distance.
>
> > In the appendix you say that there is high variability in the model configurations selected by FreSh, even within a single dataset. But how much of this variability is random vs. actually important for performance? I would be curious to see how per-image FreSh compares to a dataset-wide FreSh: how much of the advantage of FreSh comes from optimizing for each individual image, and how much can be achieved just by making an appropriate hyperparameter choice for each dataset.
>
> Due to the repeated measurement of the Wasserstein distance, the selection of embedding hyperparameters is highly consistent and not random. The question of how much of FreSh's advantage comes from optimizing for each individual image versus selecting an appropriate hyperparameter value for the entire dataset is intriguing. However, it is not clear which specific hyperparameter value from those proposed by FreSh should be chosen for a dataset-wide approach, and the answer would likely depend on the specific strategy.

---

### Official Review · Reviewer_vNDz · 2024-11-01

**Soundness:** 3
**Presentation:** 3
**Contribution:** 3
**Rating:** 8
**Confidence:** 3

**Summary:**

The authors propose a method for selecting the frequency hyperparameter involved in using periodic activations in implicit neural representations. Selection of the frequency hyperparameter is critical to INR performance, and authors often either select arbitrarily e.g. $\omega=30$, or employ a costly grid-search. The authors propose selecting this hyperparameter based on the target signal frequencies, by minimising the Wasserstein distance between the target signal spectrum and the untrained network output spectrum. This is conducted by searching across candidate frequencies. The authors demonstrate performance improvements across images, neural radiance fields, and videos. Overall this is a nicely written paper which introduces a simple method with interest to the INR community.

**Strengths:**

There are a number of strengths in this paper:
- The issue is well-motivated: selection of frequency is a common frustration when using SIREN implicit neural representations. A method for a-priori selection of frequency rather than needing to grid-search across trained networks will be useful for the community.
- The method is conceptually simple and should be easy to implement / include in INR pipelines meaning it may be broadly applicable.
- Experimental results extensively tested with results averaged across seeds for multiple modalities including image datasets (WikiArt, FFHQ, Chest X-Ray, etc), video, and neural radiance fields.
- The authors have incorporated their method in multiple pipelines (Siren, Fourier, Hashgrid embeddings, Finer). Results broadly indicate that the method leads to a performance improvements within these pipelines.
- The experiments are well ablated, with key components (e.g. spectrum size n) evaluated with multiple configurations and datasets
- The paper is well written, logically structured, and clearly presented. Existing literature is reviewed well.

**Weaknesses:**

There are a few minor weaknesses in the paper. These principally relate to small aspects of presentation (e.g. Figure 2). I've listed these in the Questions below as they are largely minor issues rather than critical weaknesses.

In terms of technical weaknesses, while the method avoids a costly grid search across trained networks it still requires a grid search across candidate frequencies on untrained networks (this cost will be negligible in comparison). In addition, the method introduces an additional hyperparameter (selection of the spectrum cut-off). If I'm reading it correctly, this hyperparameter would still need a trained network to be evaluated (even though the authors note it transfers across signals). This could reduce some of the benefit of the method, especially if the frequency hyperparameter needs to be searched in combination with the frequency cut-off. Could the authors clarify this issue?

**Questions:**

- The description of a vector spectrum by summing the DFT appears a little non-standard (L285-7). Could the authors provide a reference for this approach (potentially outside the INR literature?)
- In Appendix C it is mentioned that the for image regression the same image is resampled 10 times due to randomness of the model output. The same network should be deterministic in output - does the randomness occur due to re-sampling the network with reinitialisation?
- The paper mentions that other activations are addressed L131. Have experiments on non-periodic functions (e.g. Gaussian) been performed?
- Could the authors provide details on the compute required for Fresh on more complex signals like NeRF (L414-6 notes that this is negligible for images).
- Table 5 records video results. It mentions 'Results for NeRF are provided for reference only as it is incompatible with FreSh'. Could the authors clarify this?
 - Could the authors clarify what is meant by 'direction-dependent frequency magnitudes' and its importance for video (L510)?

Minor Presentation:
- The presentation of Figure 2 could be improved (visually the figure is difficult to interpret without the caption / main text). Tweaking this figure (perhaps additional labeling) would greatly improve the clarity of the method and extend its impact.
- The font in figures 12, 13, 14 is inconsistent
- The text size in Table 6 is inconsistent / appears to have been scaled
- Figure 5 - suggest changing 'Thousands to steps' to 'Steps ('000)'
- Figure 1 could be improved (the PSNR labels are difficult to distinguish)
- Left-hand quotation makes can be done in Latex using the ` character

---

> ### Author Response · Authors · 2024-11-21
>
> Thank you for your feedback. We fixed the minor presentation comments and addressed the rest below. We redesigned Figure 2 to improve its clarity.
>
> > The description of a vector spectrum by summing the DFT appears a little non-standard (L285-7). Could the authors provide a reference for this approach (potentially outside the INR literature?)
>
> To the best of our knowledge, there is no standard method for converting a 2-dimensional DFT into a vector. Our goal was to simplify the 2D DFT into a representation that ignores signal directionality (note that the frequency spectrum representation remains the same for a transposed image), since randomly initialized frequency embeddings do not make this distinction. Given that transposing the image is equivalent to transposing its Fourier representation, we find that summing along the diagonals works well for our purposes. Additionally, collapsing the 2D DFT into a 1D vector allows us to use an explicit formula for the Wasserstein distance, rather than relying on an approximation.
>
> If the reviewer feels it would improve clarity, we are open to renaming the "spectrum vector" to "frequency vector" in the final version.
>
>
> > In Appendix C it is mentioned that the for image regression the same image is resampled 10 times due to randomness of the model output. The same network should be deterministic in output - does the randomness occur due to re-sampling the network with reinitialisation?
>
> Yes, the randomness arises from using a **different weight initialization in each iteration**. **We updated the FreSh algorithm in the appendix to clarify** that model weights are reinitialized for each output calculation.
>
> > The paper mentions that other activations are addressed L131. Have experiments on non-periodic functions (e.g. Gaussian) been performed?
>
> Both **Finer** (Liu et al., 2024) and **Wire** (Saragadam et al., 2023) utilize non-periodic activations, with Wire being a generalization of Gaussian activations.
>
> > Could the authors provide details on the compute required for Fresh on more complex signals like NeRF (L414-6 notes that this is negligible for images).
>
> The compute required for FreSh is similar across all modalities, including NeRF.
> FreSh consists of two main steps:
> 1. Initializing INR models (10 in our case) and performing inference.
> 2. Calculating the DFT, spectrum vectors, and Wasserstein distance.
>
> Since step 1 always outputs images, the compute required for step 2 doesn't depend on modality. The compute for step 1 (10 initializations and inferences) is negligible compared to training, which involves thousands of inferences.
>
>
> > Table 5 records video results. It mentions 'Results for NeRF are provided for reference only as it is incompatible with FreSh'. Could the authors clarify this?
>
> In this context, "NeRF" refers specifically to Positional Encoding, not the NeRF task. We have updated the manuscript to use "Positional Encoding" for clarity.
>
> > Could the authors clarify what is meant by 'direction-dependent frequency magnitudes' and its importance for video (L510)?
>
> This refers to the qualitative differences in signal characteristics along different directions (e.g., temporal vs. horizontal). For example, in videos, scene transitions often cause drastic changes in pixel values between frames, a phenomenon not typically observed along the spatial dimensions. To clarify, we have added Figure 16 (page 24), which illustrates these changes. For additional discussion please refer to our general response.

---

> > ### Comment · Reviewer_vNDz · 2024-11-25
> >
> > I thank the authors for their rebuttal and have reviewed the revised draft. My questions have been addressed and I am happy to increase my score. It is encouraging to see that the method appears to be orthogonal to other initialization methods (e.g. the FAI baseline in response to swg8) and I suggest that the authors include this experiment in the supplementary materials.

---

> > > ### Author Response · Authors · 2024-11-26
> > >
> > > We sincerely thank the reviewer for considering the revised manuscript and additional experiments and for adjusting the score. We have updated the supplementary materials to include additional experiments.

---

### Author Response · Authors · 2024-11-21

We would like to thank the reviewers for their thoughtful feedback. We are encouraged that most reviewers (swg8, cowA, vNDz) found our method **easy to understand, extensively tested, and promising for wider adoption by the INR community**. Additionally, we appreciate the positive remarks about the paper being well-written (vNDz, cowA, TDK4), well-motivated (vNDz, cowA), and potentially applicable to other fields (TDK4).

We have responded to the comments of each reviewer and revised both our manuscript (with major changes marked in red) and the supplementary appendix, as outlined below.
Revised Figure 2 for improved clarity. [vNDz, TDK4]
Updated the description of the "Optimization of embedding weights" experiment (Section A, Appendix). [swg8]
Added visualizations of model outputs and spectrum vectors (Figure 8, Appendix). [swg8]
Added an analysis of video signals (Figure 16, page 24). [TDK4, swg8, vNDz]
Added InstantNGP as a baseline for video. [TDK4]
The primary concerns raised relate to (1) the impact of using time inputs in video experiments (TDK4, swg8, vNDz) and (2) the compatibility of our method with high-frequency models like NeRF (positional encoding) and Wire (TDK4, cowA). We address these issues below.


**Video experiments**

In our experiments, we use ResFields, which adjust MLP weights based on the time input. We test two configurations: (1) the MLP accesses all input coordinates, and (2) time is excluded from the input (though still used for weight modification). We observe that Fresh improves performance only in the latter configuration.

Our hypothesis is that this behavior arises from the qualitative differences between time and spatial inputs. For instance, in videos, it is more common for a pixel (e.g., background) to remain static over time than for an entire row or column of pixels to have the same value. To illustrate this, we added Figure 16 (page 24) in the appendix. However, it is also possible that removing time simplifies the task. Further investigation would be needed to fully understand this effect. The exact cause is likely a complex mix of these factors, including model capacity, stability, and others. We believe that exploring this topic further, for example by focusing on the relation between frequencies of the target signal along specific directions, could provide valuable insights and significantly advance INR research.


**High frequency models**

We first highlight the significant difference between embeddings in high-frequency models (e.g., Positional Encoding, Wire) and others (e.g., Siren, Fourier, Finer). For instance, Siren embeddings typically span 10–30 periods over the input domain, while Positional Encoding (also known as NeRF) uses at least 512 periods (considering a typical value of L > 9). To visualize this difference, we added a section in the appendix (“additional spectrum examples”), together with Figure 8 (page 19),  which compares model outputs and spectra.

Interestingly, despite the embedding’s frequency mismatch with the data, high-frequency models perform reasonably well—a phenomenon not well explained in the literature. Since their performance appears unaffected by the embedding's alignment with the data, such models do not benefit from fine-tuning their spectrum using FreSh.

It’s also worth noting the drawbacks of high-frequency embeddings, particularly instability. As highlighted by Yang et al., disabling high frequencies in NeRF can improve performance in sparse settings (see https://arxiv.org/abs/2303.07418).

---

### Meta-Review · Area_Chair_zTw5 · 2024-12-19

**Metareview:**

In this paper the authors present a method for implicit neural representation (INR) techniques which aims to avoid the loss of the high frequency content of the representation.  In particular, the authors provide a method for hyperparameter selection based on a Wasserstein metric between the target signal spectrum and the output spectrum of the untrained network.  With this initialization, the authors then demonstrate experimentally that their approach can achieve similar performance as other, much more computationally extensive, techniques that perform a grid search over model hyperparameters.

The reviewers are overall largely in consensus that this paper makes a meaningful contribution and should be accepted.  Two reviewers have concerns that the overall performance of the method does not significantly outperform baseline methods in terms of raw task performance; however, the authors note that this performance is achieved from their simple hyperparameter initialization scheme which eliminates the need for a significantly more costly grid search.  I find this to be a convincing argument and agree with the general consensus of the reviewers to accept the paper for publication.

**Additional Comments On Reviewer Discussion:**

The authors were generally responsive to the reviewers during the discussion period, with several reviewers increasing their scores following the rebuttal period (in some cases significantly so).

---

### Decision · Program_Chairs · 2025-01-22

Accept (Poster)